



# Quantifying the spatial extent and intensity of recent extreme drought events in the Amazon rainforest and their impacts on the carbon cycle

Phillip Papastefanou[1], Christian S. Zang[1], Zlatan Angelov[1], Aline Anderson de Castro[2], Juan Carlos Jimenez[3], Luiz Felipe
Campos De Rezende[2], Romina Ruscica[4,5,6], Boris Sakschewski[7], Anna Sörensson[4,5,6], Kirsten Thonicke[7], Carolina Vera[4,5,6],
Nicolas Viovy[8], Celso Von Randow[2] and Anja Rammig[1]

[1] Technical University of Munich, TUM School of Life Sciences Weihenstephan, Freising, Germany

[2] Earth System Sciences Centre, National Institute for Spatial Research, São José dos Campos, São Paulo, Brazil

[3] GCU/IPL, University of Valencia, Valencia. Spain.

[4] Universidad de Buenos Aires, Facultad de Ciencias Exactas y Naturales, Departamento de Ciencias de la Atmósfera y los
Océanos. Buenos Aires, Argentina.

[5] CONICET – Universidad de Buenos Aires. Centro de Investigaciones del Mar y la Atmósfera (CIMA). Buenos Aires,
Argentina.

[6] CNRS – IRD – CONICET – UBA. Instituto Franco-Argentino para el Estudio del Clima y sus Impactos (UMI 3351
IFAECI). Centro de Investigaciones del Mar y la Atmósfera (CIMA). Buenos Aires, Argentina.

[7] Potsdam Institute for Climate Impact Research (PIK), Telegraphenberg A31, Potsdam, 14473, Germany

[8] LSCE, CEA-CNRS-Univ Paris-Saclay, Saclay, France

*Correspondence to*: Phillip Papastefanou (papa@tum.de)



**Abstract.** Over the last decades, the Amazon rainforest was hit by multiple severe drought events. Here we assess the severity and spatial extent of the extreme drought years 2005, 2010, and 2015/2016 in the Amazon region and their impacts on the carbon cycle. As an indicator of drought stress in the Amazon rainforest, we use the widely applied maximum
cumulative water deficit ($\Delta$MCWD). Evaluating an ensemble of ten state-of-the-art precipitation datasets for the Amazon region, we find that the spatial extent of the drought in 2005 ranges from 2.8 to 4.2 (mean = 3.2) million km² (46 – 71% of the Amazon basin, mean = 53%) where $\Delta$MCWD indicates at least moderate drought conditions ($\Delta$MCWD anomaly < 25 mm). In 2010, the affected area was about 16% larger, ranging from 3.1 up to 4.6 (mean = 3.7) million km² (52 – 78%, mean = 63%). In 2016, the mean area affected by drought stress was similar to 2005 (mean = 3.2 million km²; 55% of the Amazon
basin), but the general disagreement between data sets was larger, ranging from 2.4 up to 4.1 million km² (40–70%). In addition, we compare differences and similarities among datasets using the self-calibrating Palmer Drought Severity Index (scPDSI) and a rainfall anomaly index (RAI). We find that scPDSI shows a much stronger, and RAI a much weaker drought impact in terms of extent and severity for 2016 compared to $\Delta$MCWD. Using an empirical $\Delta$MCWD-mortality relationship, we calculate biomass losses of the three drought events. We show that eight of ten datasets agree on biomass losses of about
1.8 PgC for the drought years 2005 and 2010, indicating that the more intense drought in 2005 equals a larger total area of the 2010 drought regarding biomass loss. For the 2015/2016 drought event, datasets show a large variability of biomass loss induced by drought stress ranging from 1.3 to 2.7 PgC with a mean loss of 1.8 PgC. Disagreement across datasets increased, (1) when comparing the total area of more severe and extreme drought signals and (2) when comparing spatial drought location across datasets. Generally, only half of the datasets agreed on the location of a drought event. We conclude that for
deriving impacts of droughts to the Amazon Basin based on precipitation, an ensemble of datasets should be considered. This is especially relevant when assessing the impact of drought on the Amazon rainforest and its carbon cycle.



## 1 Introduction

The severe drought events occurring in 2005, 2010 and 2015/16 in the Amazon basin are reasons for concern regarding their
frequency and severity, and their impacts on the Amazon rainforest. Different large-scale atmospheric processes related to
increased sea surface temperature (SST) in the Pacific and the Atlantic Ocean seem to be responsible for such repeated
mega-drought events (Coelho et al., 2012): While the drought 2015/16 was driven by a record-level El Niño event enhanced
by the strong underlying global warming trend (Jimenez et al., 2018), the 2010 drought was a combination of a moderate El
Niño event and anomalously warm SSTs in the tropical North Atlantic (Marengo & Espinoza, 2016; Marengo et al., 2011).
Similarly, the 2005 drought was attributed to anomalies of warm SSTs in the North Atlantic (Marengo et al., 2008; Zeng et
al., 2008). In consequence, such events differ in their strength, their timing and spatial patterns, and thus, impacted regions
differ. While drought events related to El Niño events show a Southwest to Northeast gradient with dry conditions over the
NE Amazon region (Malhi et al., 2008), drought events caused by anomalously warm North Atlantic SSTs show a North-
South gradient with dry conditions in the southern Amazon region (Lewis et al., 2011; Marengo et al., 2008). Even in the
case of El Niño events, SSTs anomalies over the Eastern Pacific (EP) or the Central Pacific (CP) can lead to different
impacts and spatial patterns of drought (Jimenez et al., 2019). In addition to their influence on temperature, recent El Niño
events also showed amplified atmospheric vapor pressure deficit anomalies (Barkhordarian et al., 2019; Rifai et al., 2019).
The impacts of such drought events on humid tropical forests, which are often not adapted to longer-lasting dryness, are
severe. Increased forest mortality connected to drought events was observed in central and southern Amazonia (Lewis et al.,
2011; Phillips et al., 2009), as well as shifts in tree species composition (Esquivel-Muelbert et al., 2019). Droughts are
assumed to be one of the main drivers for the observed decline in the Amazon carbon sink, indicating that more carbon is
lost to the atmosphere than taken up by the forest (Hubau et al., 2020). Thus, such extreme drought events are altering the
carbon cycle of the Amazon forest already today (Gloor et al., 2015; Hubau et al., 2020; Phillips et al., 2009).

Losing tropical forests in the Amazon region through increased mortality under drought also has implications for regional
and continental scale water cycling (Ruiz-Vásquez et al., 2020). The rainforest transpires enormous amounts of water which
is transported by winds to remote regions far beyond the borders of the rainforest (e.g. Dirmeyer et al., 2009; van der Ent et
al., 2010; D. C. Zemp et al., 2014; Zemp et al., 2017a). In addition, the ongoing deforestation in the Amazon rainforest
further decreases forest cover and thus, transpiration rates, leading to a rainfall decline and enhanced drought conditions in a
positive feedback loop (Miralles et al., 2019; D. C. Zemp et al., 2017a; Zemp et al., 2017b). It can be expected that ongoing
climate change most likely will cause stronger and more frequent drought events in the Amazon (Cai et al., 2015; Jia et al.,
2019; Marengo & Espinoza, 2016).

For assessing the severity, the spatial extent and, in particular, the impacts of such drought events on existing ecosystems,
different gridded precipitation datasets are available which in some cases differ strongly in magnitude and spatio-temporal
distribution of precipitation amounts (Golian et al., 2019). Typical problems of precipitation data for South America
encompass the underestimation of extreme rainfall events in both dry or wet seasons (Blacutt et al., 2015; Giles et al., 2020).



Therefore, while for the Amazon region, the recent drought events have been assessed in terms of severity (Jiménez-Muñoz et al., 2016; Jimenez et al., 2018) and impacts (Phillips et al. 2009, Lewis et al. 2011) based on single precipitation data sets, a systematic analysis of how the most frequent used precipitation datasets differ regarding the spatial extent, location and severity of recent extreme drought events, is currently missing.

For our study, we selected ten precipitation datasets: (1, 2) Data from the Tropical Rainfall Measurement Mission (TRMM) version 6 and 7 (Huffman et al., 2007) which have been frequently used, e.g. to estimate drought impacts on the carbon balance (Lewis et al., 2011; Malhi et al., 2009) and are assumed to represent precipitation patterns in the Amazon region best since they are derived from radar measurements (Huffman et al., 2007). (3) CHIRPS (Climate Hazards group Infrared Precipitation with Stations, Espinoza et al., 2019), which has been used to study regional hydro-climatic and environmental

changes in the Amazon Basin. These two datasets only provide precipitation and no information about other climatic variables such as temperature or radiation. In addition, we selected five datasets that are often used as drivers for ecosystem models (e.g. in Forkel et al., 2019; Yang et al., 2015) and – in contrast to the other datasets – provide information about other climate variables: Data from the Climate Research Unit (CRU) with a joint project reanalysis (NCEP, National Centers for Environmental Prediction) applied, (4) the CRUNCEP (version 8, Viovy, 2018), (5) the WATCH-WFDEI  (WATCH: Water

and Global Change, Weedon et al., 2011. WFDEI: WATCH Forcing Data methodology applied to ERA-Interim, Weedon et al., 2014) dataset, originally derived from global sub-daily observations merged with integrations from a general circulation model, (6) the GSWP3 (Global Soil Wetness phase 3, Kim et al. in prep) dataset which is closely related to WATCH-WFDEI, relying on a similar forcing but with a different bias correction applied, (7) the newer GLDAS s(Global Land Data Assimilation System) 2.1. which is derived from various geostationary infrared satellite measurements and microwave

observations (Rodell et al., 2004), (8) the ERA-Interim dataset which is generated using a forecast model driven with different input datasets (Dee et al., 2011), (9) the latest ECMWF atmospheric reanalysis dataset, ERA5, which is the successor of ERA-Interim, providing higher spatial and temporal resolutions and a more recent model and data assimilation system than the previous ERA-Interim reanalysis (Albergel et al., 2018), and, finally, (10) the GPCC (named after the Global Precipitation Climatology Centre) dataset (Schneider et al., 2018), which is based on globally available land stations (rain

gauges) combined with an empirical interpolation method (Willmott et al., 1985). A more detailed description of the datasets is given in the methods section.

We evaluate the precipitation datasets based on the Maximum Cumulative Water Deficit (MCWD; Aragão et al., 2007), a well-established drought index that is particularly suitable for estimating drought stress in the Amazon region (e.g. Esquivel-Muelbert et al., 2019; Lewis et al., 2011; Y. Malhi et al., 2009; Phillips et al., 2009; Zang et al., 2020). In addition, we

included two other measures to complement our analysis: Rainfall anomaly index (RAI), which does account for the mean deviation (in units of standard deviation) of precipitation during the driest months of the year and scPDSI (self-calibrating Palmer Drought Index, Wells et al., 2004). scPDSI has a more complex formulation compared to RAI and MCWD and takes available soil water content into account. Both RAI and scPDSI have been used in studies describing the recent Amazonian drought events (e.g. Jiménez-Muñoz et al., 2016; Lewis et al., 2011).





The goals of our study are (1) to analyze and quantify the uncertainty in drought strength, extent and location of three recent
Amazon droughts in the years 2005, 2010 and 2015/2016 in ten state-of-the-art precipitation datasets based on MCWD; (2)
to examine differences among these drought events by taking two additional drought indicators RAI and scPDSI into
account; and (3) give an estimate of the impacts of the three drought events on the carbon cycle by estimating potential
biomass losses.




## 2 Methods

### 2.1 Study area

Our study covers the Amazon river basin as delineated by Döll & Lehner (2002, see black contour in Fig. 1). Using 0.5°
spatial resolution in longitude and latitude results in 1946 grid cells of interest for this study area. To compare spatial
differences of drought extent in more detail, we subdivided the Amazon Basin into 13 regions based on countries and
Brazilian states intersecting with the area (SI Fig. 1). Note that differences in the comparison of our results with Lewis et al.
(2011) arise because of differences in the delineation of the Amazon region, i.e. the area used in our study is 0.6 Mio km²
larger.

### 2.2 Data sources

In the following, we briefly describe the ten precipitation datasets applied in our study (see also Table 1): The Tropical
Rainfall Measuring Mission (TRMM v7) product (Huffman et al., 2007) is a precipitation-only dataset based on multiple
microwave-infrared satellite data developed as a joint product between NASA and the Japan Aerospace Exploration Agency
(JAXA). We also included the predecessor v6 for comparison in our study, because it has been frequently and prominently
used to derive drought impacts to the Amazon Basin (e.g. Lewis et al., 2011; Phillips et al., 2009) and shows significantly
lower precipitation throughout the basin compared to v7 (Seto et al., 2011). Both TRMM datasets are from now on denoted
as TR6 and TR7. CHIRPS (Climate Hazards group Infrared Precipitation with Station) is a novel dataset (Funk et al., 2015
from now on denoted CHR) which is a quasi-global (full longitude, but only 50°S – 50°N latitude extent) precipitation-only
merged product, based on multi-satellite estimates (similar to TR6 and TR7) and approx. 2,000 in-situ observations per
month in South America. TR6, TR7 and CHR share the quasi-global spatial extent, however, in comparison to TR6 and TR7
with a resolution of 0.25° x 0.25°, CHR has a much higher spatial resolution of 0.05° x 0.05°. ERA-Interim (from now on
denoted as ERI) is an atmospheric model that assimilates observation-based estimates from the GPCP-dataset (Adler et al.,
2003) of the atmosphere during runtime (Dee et al., 2011). Although ERI might show some anomalies in tropical biomes (Di
Giuseppe et al., 2013), it has been used for drought evaluation of the Amazon rainforest (Jiménez-Muñoz et al., 2016) and
also as a forcing dataset for dynamic vegetation models (DVMs; e.g. Maignan et al., 2011; Poulter et al., 2011). ERA5
(Muñoz-Sabater et al., 2018), from now on denoted as ER5, shows improvements in, e.g., land evapotranspiration, surface
soil moisture and turbulent heat fluxes over its predecessor ERI (Albergel et al., 2018). Similarly, CRUNCEP (Viovy, 2018
from now on denoted as CRU) is generated based on a reanalysis from the national centers for environmental prediction
(NCEP) and the National Center for Atmospheric Research (NCAR), corrected with the CRU TS3.2 (Harris et al., 2014)
dataset. GPCC (from now on denoted as GPC) is mainly based on data from rain gauge land stations. Similar to CRU, it is
also based on a reanalysis and has been used in global drought studies (Ziese et al., 2014). Both GPC and CRU cover the
longest periods of all selected datasets in this study with timespans from 1891 until 2016 and from 1901 until 2016,
respectively. WATCH-WFDEI (denoted as WAT from now on; Weedon et al., 2011; 2014) is based on the reanalysis ERI



corrected with GPC precipitation. GSWP3 (Kim et al. in prep; from now on denoted as GSW) is based on the atmospheric reanalysis method "20CR" (20th Century Reanalysis version 2, Compo et al., 2013), which has been dynamically

downscaled to 0.5° x 0.5° resolution. Corrections with observational data have not only been applied to precipitation but also to short/longwave radiation, air temperature and the daily temperature range. Both WAT and GSW end in the year 2010. The GLDAS 2.1 (from now on denoted as GLD) dataset is built by using the 'Noah Land surface model' forced by the Goddard Earth Observing System (GEOS) Data Assimilation System with corrected precipitation and radiation (Rodell et al., 2004; Sheffield et al., 2006). Starting in January 2000 (Version 2.1), it is the dataset with the latest time onset and hence defines

the lower-bound time interval considered in this study. For the 2015/2016 drought event, only seven datasets were available as three of the datasets (TR6, GSW and WAT) end before. All datasets were (if not directly available) converted to 0.5° x 0.5° spatial resolution and to monthly time steps.

### 2.3. Drought indices and evaluation of drought area and extent

2.3.1 Calculation of maximum climatological water deficit (MCWD)

We calculate MCWD based on Aragão et al. (2007) defining water deficit (WD) as follows:

$$WD(t) = P(t) - ET(t), \tag{1}$$

where $WD(t)$ stands for water deficit, which is calculated for a time step t, in this case for a monthly time step, $P(t)$ for monthly precipitation and $ET(t)$ for monthly evapotranspiration. To estimate the impacts of persistent drought events, the cumulative water deficit ($CWD$) is defined as the accumulation of water deficit of each month of the hydrological year (see

below for details) for which $P(t)$ is smaller than $ET(t)$, hence $WD(t)$ is negative. MCWD is the most negative value of $CWD(t)$ over a specific period. For a complete mathematical definition, see Supporting Information Methods S1. As proposed by Aragão et al. (2007), we use a fixed value for $ET(t) = ET_{fixed} = 100$ mm month$^{-1}$ derived from ground measurements of evapotranspiration in different locations and seasons in Amazonia (von Randow et al., 2004; da Rocha et al., 2004). As a result, water deficit builds up whenever the monthly rainfall $P(t)$ falls below 100 mm.

We calculate annual MCWD for the hydrological year from October of the previous year to September of the succeeding year, e.g. the MCWD for the year 2000 is calculated from October 1999 to September 2000 (similar to Lewis et al., 2011). Similarly, for deriving the drought severity, we calculated the MCWD anomaly (ΔMCWD) for 2005 and 2010 by first calculating the mean MCWD for the "baseline" period from 2000 to 2010, thereby excluding the years 2005 and 2010. To derive ΔMCWD, the baseline period is subtracted from the mean value of 2005 and 2010, respectively. The same procedure

was applied for calculating ΔMCWD for 2016, extending the baseline period to from 2000 to 2016 and additionally excluding the year 2016. We excluded the drought years from the baseline period as the high proportion of drought years would bias the mean water stress (Lewis et al., 2011). We investigated also the effect of including drought years in the baseline calculation and the role of a longer baseline period (Fig. S1). Similar to Lewis et al. 2011, we defined ΔMCWD <




−25 mm as moderate drought stress because at this level, tree mortality already significantly increased in their inventory
plots. We further defined $\Delta MCWD < -100$ mm as severe and $\Delta MCWD < -150$ mm as extreme drought stress.

### 2.3.2. Calculation of rainfall anomaly index (RAI)

For the rainfall anomaly index, dry season rainfall was taken as the mean precipitation from July-September following Lewis
et al. (2011). For each year, the 'standardized anomaly' was calculated as the anomaly of rainfall expressed as the difference
in units of standard deviation from the mean dry season rainfall over all years. Like for to the MCWD calculation, we
excluded the drought years 2005 and 2010 from the mean dry season precipitation calculation from a baseline period 2000-
2010 to investigate the drought impacts of 2005 and 2010, and for 2016 we selected a baseline period from 2000 to 2016
excluding 2005, 2010 and 2016. We defined $RAI < -1$ to represent moderate, $RAI < -2$ to represent severe, and $RAI <
-3$ to represent extreme drought stress.

190

### 2.3.3. Calculation of the self-calibrating Palmer Drought Severity Index (scPDSI)

The self-calibrating Palmer Drought Severity Index (scPDSI, Wells et al., 2004) has in recent studies been used to assess the
impacts of droughts on the Amazon basin (e.g. Jiménez-Muñoz et al., 2016). It improves the original PDSI by using a self-
calibrating procedure based on historical climate data, eliminating the empirically derived climatic characteristics. Next to
195    precipitation, it also takes monthly potential evapotranspiration ET into account. In our study, we use ET data generated by
the ER5 reanalysis. Additionally, the scPDSI takes soil water capacity as input, which we assumed here as a constant value
of 100 mm. scPDSI was estimated using the *R* package *scPDSI* (Ruida et al., 2018).

To enable comparison with the MCWD and RAI, we selected identical baseline periods from 2000 to 2010 for the 2005 and
2010 events and from 2000 to 2016 for the 2016 drought event. We also adopted the categorization from Jiménez-Muñoz et
al. (2016) and Wells et al. (2004) with $scPDSI < -2$ representing moderate, $scPDSI < -3$ severe and $scPDSI < -4$
extreme drought stress.

### 2.3. Calculation of drought area and extent

Each grid cell's area was approximated as a trapezoid to its boundary coordinates (in 0.5° x 0.5° resolution), resulting in an
area between 2900 and 3090 km² per grid cell. Accumulating the associated areas over all grid cells resulted in a total area of
5.94 million km² representing the Amazon Basin. Note that for comparison of our results with Lewis et al. (2011)
differences in absolute areas arise because of differences in study area size (5.94 vs. 5.3 million km², respectively). For the
calculation of the drought-affected area, we summed up the area of grid cells that matched the respective drought
classification (e.g. $\Delta MCWD < -150$ mm for extreme drought stress). The spatial agreement of drought location among
datasets was estimated by selecting the grid cells matching the drought classification per dataset and subsequently counting
the number of datasets per grid cells showing the respective drought classification.



### 2.4. Estimating carbon losses during drought events

To estimate carbon loss during drought events, we used a simple linear relation between MCWD and carbon losses in the

Amazon basin derived from plot measurements (Lewis et al., 2011):

$$\Delta AGB = 0.3778 - 0.052 * \Delta MCWD \tag{2}$$

Here, ΔAGB denotes the change in aboveground biomass, i.e. biomass carbon losses. The equation was derived from

Amazon plot inventory data measured across the RAINFOR network to estimate the impact of the 2005 drought event
(Lewis et al. 2011). To calculate ΔAGB in Eq. 2, we used the ΔMCWD of each gridcell for each drought year calculated for
each of the precipitation datasets in our study. The total biomass carbon loss (in Pg C) across the Amazon basin is then
calculated by summing up ΔAGB for all gridcells weighted by each gridcell's size.




## 3. Results

All areas in the following section are expressed as percentage with respect to the entire Amazon basin according to our delineation (5.94 million km²). For an overview of the areas affected in million km², see Table 2 and 3.

### 3.1 Comparison of total drought area based on ΔMCDW

We first evaluate differences in the two TRMM products, TR6 and TR7. For 2005 and 2010, we find similar spatial patterns for TR7, as in Lewis et al. 2011 for TR6 (Fig. 1a, b). Regarding drought intensities, TR7 agrees with its predecessor TR6 for 2005, showing a slightly smaller area (4% less), but an 11% smaller area for 2010. ΔMCWD calculated from TR7 indicates that the North-Western region of the Amazon Basin (particularly the Roraima region) was hit extremely by drought stress in 2016 with 7% of the area having $\Delta\text{MCWD} < -150$mm (Fig. 1c). Furthermore, in 2016 about 15% of the basin was severely

affected by drought stress located at the Western part and scattered in South-Eastern Amazonia. Moderate drought stress was found throughout 54% of the basin also affecting central and western Amazonia (Fig 1c).

Across all precipitation datasets, in 2005, an area ranging from 46 to 71% (mean 53%) of the whole Amazon basin was moderately affected (Table 2, Fig. 2a). GSW and GLD displayed the smallest area affected by moderate drought (2.6 million

km², Tab. 1, Fig. 2), while ER5 showed a vast affected area (4.2 million km²), an area about 12% larger than displayed by ERI. For severe and extreme drought conditions, CHR shows the smallest affected area with 6% of the basin and no affected area, respectively. For severe drought conditions, CRU suggests that approximately 16% more of the basin area was affected in comparison to CHR (1.6 million km² vs. 0.4 million km²). CRU also encompasses the largest area of extreme drought stress (0.7 million km²; 12% of the basin less than $\Delta\text{MCWD} < -150$mm).

During the 2010 drought, a larger area was affected by moderate drought ranging between a minimum of 52% (GPC) and a maximum of 76% (TR6), which is about 10% larger than during the 2005 drought (3.1 million km² vs. 4.6 million km², Tab. 2, Fig. 2). In addition, the area with severe drought extent was on average 3% larger compared to 2005. The area affected by extreme drought was smaller than during the 2005 drought. Particularly, ER5 and TR6 showed the largest area affected throughout the three drought classifications (Fig. 2b).


For 2016, two datasets (CHR and CRU) showed with 40% a considerably smaller area that was moderately affected by drought compared to ER5 and ERI with 69% and 63% of the area affected, respectively (datasets ranging between 2.4 and 4.1 million km²). Generally, in 2016, the size of the area affected by moderate drought was in between the size of the area affected 2005 and 2010, but the extent of severely and extremely drought-affected areas was larger. Here, particularly ERI

(closely followed by ER5) showed the largest affected area, with 30% severely affected and 18% extremely affected.





### 3.2 Spatial agreement of rainfall datasets using ΔMCDW

While the agreement of total area affected by drought is relatively high (see 3.1), the data sets are only partly in agreement regarding the spatial patterns and locations of the 2005, 2010 and 2016 droughts (Fig. 3). For 2005, all datasets are in agreement regarding the drought epicenter being located in Central Amazonia mainly affecting the Brazilian states

*Amazonas* and *Acre* (Fig. S4 b, d). All ten datasets also agree that an area of about 15 % of the Amazon Basin was at least moderately affected (Fig. 3a). Only a small overlap was found for the area affected by severe and extreme drought stress (Fig. 3b, c). Here, only half of the datasets agreed on 11% of central Amazonia being severely and 4% extremely affected.

For 2010, all datasets agreed on an affected area of 11% in the Amazon basin, and half of the datasets agreed on an area of 72% of the Amazon Basin being moderately affected by drought stress (Fig. 3d). The 2010 drought displayed no central

hotspot, but three most affected areas in the Eastern, Southern and central part of Amazonia on which most of the datasets agreed (Fig. 3d). Severe drought stress in 2010 was located in the southern part of Amazonia, where four datasets agreed (Fig. 3e), while for extreme drought stress almost no overlap between datasets was found (Fig. 3f).

For 2016, all datasets agreed on an area of about 8% for moderate drought stress and half of the datasets agreed on 54% of the basin being affected (Fig. 3g). Agreement for severe and extreme drought stress was higher compared to the other

drought years (Fig. 3h, i). Most of the data sets located the epicenter of the drought in the North-Western part of Amazonia. Some datasets also showed the South-Central part of the basin being severely affected (Fig 3i).

### 3.3 Estimating the variation of carbon losses during drought events

For the different precipitation datasets and based on the linear relation between ΔMCWD and ΔAGB, we derive carbon losses for 2005 to be in the range of 1.3-1.9 Pg C with CHR showing the smallest and CRU the strongest impact regarding

carbon losses (Fig. 4). The mean biomass loss over all datasets was 1.6 Pg C with six of the ten estimates from the different datasets being close to that mean (difference of ΔAGB less than 0.15 PgC to the mean value). For 2010, carbon losses range from 1.5 to 2.3 Pg C with WAT showing the smallest and TR6 strongest response. Next TR6 also ER5 shows a very strong drought impact with 2.3 PgC. All other datasets show much smaller impacts between 1.6 and 1.8 Pg C comparable to the 2005 drought impact. The 2016 drought event shows the widest range of biomass loss across datasets ranging from 1.3 PgC

to 2.5 PgC. The disagreement between datasets is also larger for 2016 compared to 2005 and 2010: Both, CRU and CHR show a low impact of 1.3 Pg C, TR7 and GPC show 1.7 Pg C biomass loss comparable to the averages of 2005 and 2010. GLD, ER5, and ERI show very strong impacts of 2.1, 2.3 and 2.6 Pg C, respectively.

### 3.4 Comparison of drought indices: ΔMCDW, scPDSI and RAI

Similar to ΔMCWD, there is variable agreement among datasets when evaluating the other two drought metrics, RAI and

scPDSI (Fig. 5). scPDSI showed the lowest agreement across datasets, with mainly two datasets in agreement on areas



affected by drought for 2005. Regarding the total area affected in 2005, TR7 showed the largest area (48% of the Amazon basin, 2.8 million km²) and GLD (32%, 1.9 million km², Table 2) the smallest area affected by drought stress. Severe drought-stressed areas ranged between 16% (GLD) and 26% (CRU) and extreme drought stress between 1% (GLD) and 5% (CRU) of the basin affected. The largest rainfall anomaly (RAI) for moderate drought stress in 2005 was displayed by CHR

with 52% (3.1 million km², Table 2), followed by ER5 with 49% of the area affected. CRU showed with 29% the smallest area affected by drought stress. The area of severe drought stress was smaller using RAI compared to scPDSI, ranging from 9 to 20%. In general, the datasets displayed a more spatially connected area in the center of the Amazon basin when using RAI compared to scPDSI. RAI and MCWD agreed on the spatial location of the drought, while scPDSI showed severe drought stress in a different region (Fig 5a, d, g): Some areas showed a strong disagreement between drought indices, e.g. a

small area in Western Brazil and Peru was hit by severe drought stress according to $\Delta$MCWD and RAI (with all climate datasets in agreement). In contrast, scPDSI did not indicate abnormally dry conditions there.

In 2010, the total droughted area was similar for scPDSI and smaller for RAI compared to MCWD regarding severe drought stress (Fig. 5b, e, h): For scPDSI, in particular, GLD showed a large area of 50% of the basin severely affected (2.9 million km², Table 2), followed by CRU showing 33% affected using scPDSI. The agreement between datasets was lower compared

to the 2005 drought for both RAI and scPDSI. $\Delta$MCWD and scPDSI showed similar areas in the southern Amazon Basin severely affected by drought. According to RAI, datasets agreed on the severely affected area in the North-Western part of Amazonia, diverting from the other indices (Fig 5h).

For 2016, scPDSI showed the largest area affected by drought stress with GLD showing 62% (followed by TR7, 52%) of the basin being severely affected. Four datasets agreed on the affected area in the northeastern part of the basin (Fig. 5f). Hardly

any drought stress was visible in 2016 when calculating rainfall anomalies (RAI, Fig 5i), indicating no pronounced anomalies in dry season rainfall. Only GLD diverted from the other datasets showing 30% of the area under severe drought stress, while all other datasets found between 0-1% of the area to be affected (Table 3). $\Delta$MCWD and scPDSI again agreed on the spatial extent of the droughted area (Fig. 5c, f). Generally, scPDSI showed a much larger area severely affected by drought stress over $\Delta$MCWD and RAI.

Seasonal patterns of median $\Delta$MCWD across the Amazon basin were consistent for 2005, where all datasets showed a sudden drought impact (decline in $\Delta$MCWD) from July onwards. Only ERI and ER5 displayed a small decline already in the months before July. The 2010 drought followed similar patterns regarding $\Delta$MCWD, with a lower absolute impact (Fig 6b). For 2015, datasets agreed on a small decline in $\Delta$MCWD followed by a more substantial impact in 2016 with fewer datasets in agreement (Fig 6c). Datasets agreed well according to the seasonal patterns of scPDSI for 2005 and 2010 (Fig 6d, e). This

agreement was lower for the year 2016, in which CRU, GLD and TR7 indicate drought stress already starting in January, and ERI and ER5 only starting in September (Fig. 6f). All datasets showed a period of drought stress for longer than 12 months. Datasets generally agreed on rainfall anomaly (RAI) patterns for all of the drought years 2005, 2010, and 2016 (Fig.





6g, h, i). For 2005 the difference in rainfall was highest in June-July and for 2010 in March, August and September. The 2015/2016 drought event showed a long period of strong (negative) rainfall anomaly from August 2015 to July 2016 (Fig. 6i).



## 4. Discussion

We assessed the severity and spatial extent of the extreme drought years 2005, 2010, and 2015/2016 in the Amazon region and their impacts on the carbon cycle. When analyzing drought representation in ten different precipitation datasets for the
Amazon basin, we find that while the datasets mostly agree on the extent of the drought area, they differ in their location of drought. We show that biomass losses during 2005 and 2010 were about 1.8 PgC, indicating that the more intense drought in 2005 equals a larger total area of the 2010 drought regarding biomass loss. In 2015/2016, we find a large variability of biomass losses depending on the precipitation dataset used, ranging from 1.3 to 2.7 PgC.

**Critical aspects regarding the detection of drought events in the Amazon basin**

*Drought indices*

MCWD is one of the most widely used measures to assess drought stress in tropical forests (e.g. Lewis et al., 2011, Phillips et al., 2009, Esquivel-Muelbert et al., 2019). The calculation of MCWD only requires precipitation data and assumes a constant evapotranspiration (ET) rate of 100 mm month$^{-1}$ (Aragão et al., 2007). Although the simplicity of ΔMCWD is a
main advantage, a fixed ET (which we also used in our study) is inappropriate for regions other than the lowland tropics, where the lower supply of energy may result in lower ET values. Most importantly, an approximated ET does not account for either seasonal variation (driven mainly by radiation, temperature and phenology) or spatial variation in ET related to soil and root properties (Malhi et al., 2009). Hence, changes in ΔMCWD are purely accounting for changes in rainfall (Phillips et al., 2009). In contrast, scPDSI is driven with spatially and temporally resolved evapotranspiration data (here: ER5).
However, currently available evapotranspiration products for the Amazon rainforest show significant differences in areas and extent of evapotranspiration (Sörensson and Ruscica, 2018), hence introducing another source of uncertainty when using it for the calculation of drought indices.

The key difference between the three drought indices applied in our study is the temporal resolution: RAI is only calculated for the three driest months (July-September) and thus, for example, a rainy season with deficient rainfall is not captured.
ΔMCWD, in contrast, accumulates over 12 months and is reset to zero at the end of the hydrological year. In this way, drought events caused by low precipitation in both dry- and rainy season are captured, however, drought events lasting for longer than a year are not detected. scPDSI is not reset to zero at the end of the hydrological year and is thus captures also multi-year drought events. As an example, the 2015/2016 drought event is classified as a severe multi-year drought according to Yang et al. (2018), which is also displayed in our analysis when using scPDSI (all datasets in agreement that
more than 30% of the area were affected, Tab. 3). ΔMCWD and RAI, however, do not agree on a spatially and temporally extensive drought event (Fig. 5c, f, g, Tab. 3), but instead display distinct regions of severe drought stress. Thus, this drought event seemed not to be characterized by particularly low dry-season precipitation, but by low precipitation accumulated over a longer time period. scPDSI and ΔMCWD roughly agreed on spatial extent but scPDSI showed a more substantial drought impact indicating that precipitation levels might have been already lower than usual during the years before the 2016 drought





event happened, indicating a multi-year drought event (Yang et al., 2018). Seasonal patterns of the three drought indices
support this assumption (Fig. 6): Resetting MCWD once per year neglects any influences of drought events of the preceding
year (Fig. 6c).

A common drawback of all drought metrics used in our study is their incapability to explicitly represent the effect of
increasing atmospheric vapor pressure deficit (VPD) on plant water stress. A steady amplification of atmospheric vapor
pressure deficit (VPD) has been detected over the Amazon basin (Barkhordarian et al., 2019; Rifai et al., 2019). Such
stronger atmospheric water demand leads to additional water loss of plants during drought, subsequently increasing the
severity of droughts. Hence, the role of VPD during drought and as a driver for plant stress should not be underestimated
(Grossiord et al., 2020). With increasing data availability and better estimates of VPD across the Amazon region, it should
be included in future drought assessments (e.g. Castro et al., 2020). Furthermore, in the last decade, new methods have been
developed that assess impacts of drought on ecosystems, e.g. analyses based on solar-induced fluorescence (SIF) data show
that tall forests are less sensitive to rainfall compared to short forests (Giardina et al., 2018). Also, vegetation optical depth
(VOD) used as a proxy for water content in forests is a promising satellite-derived indicator for mortality and impacts of
droughts to forests (Rao et al., 2019). However, conducting analyses over the Amazon rainforest based on VOD is difficult,
because VOD data across tropical regions is often noisy as the high cloud cover over the rainforests generates erroneous
signals (Konings and Gentine, 2017). Future studies should estimate the impacts of droughts based on multiple drought
characteristics, e.g. Toomey et al. (2011) show that considering both, heat stress and soil moisture stress greatly improves the
explanatory power of drought impacts in the Amazon basin.

*Precipitation datasets*

For the three drought events in 2005, 2010 and 2016, ERI and ER5 diverted the most from the other datasets regarding the
size of the area affected by drought. Especially ER5 shows the largest area of moderate drought stress during all three
drought events (Fig. 2). Although TR7 and CHR are based on the same satellite data as the input, they differ regarding the
size of the drought area, especially during 2016 (Fig. 2). Lewis et al. (2011) estimated an area of 47% (2.5 million km²) of
the Amazon basin moderately affected in 2005 using the TR6 dataset, which compares well with the size of the affected area
in the GLD, GPC, and GSW datasets analysed in our study (considering our 0.6 million km² larger study area; see Methods).
For 2010, Lewis et al. (2011) reported an area of 3.2 million km² being affected in comparison to 4.5 million km² in our
analysis using TR6 with very similar spatial patterns. The newer TRMM product, TR7, however, shows less frequent rainfall
but heavier rainfall than CHR maintaining a similar total amount (Giles et al., 2020). Also, both TRMM versions (TR6 and
TR7) differ regarding the total area affected by drought in 2005 and in particular in 2010 with TR6 showing a 14% larger
area of the Amazon basin affected in our analysis. This can be explained by the generally higher precipitation rates detected
in the TR7 dataset in comparison to TR6 (Seto et al., 2011) leading to lower absolute values of ΔMCWD. Spatially, this
difference was most pronounced in the western and northern parts of Amazonia, in the *Acre* and *Roraima* states, and in Peru.
Because of such higher precipitation rates in TR7 as compared to TR6, and subsequently the much stronger drought response





according to our analysis, studies only based on TR6 might overstate the actual drought conditions and should be revisited.
Precipitation datasets usually show remarkable differences in the representation of occurrence, frequency, intensity and
location of events, mainly due to their nature of high spatial and temporal variability (Covey et al., 2016; Dirmeyer et al.,
2012). Generally, the sparse network of observations in the Amazon rainforest may explain the differences across
precipitation datasets and drought indices for datasets that rely on station data. Within the last decade, the number of
observations increased, due to a new denser network of stations. This may improve the reanalysis models that are used for
several precipitation datasets applied here, however, it does not improve datasets that only rely on gauge observations.
According to Jiménez-Muñoz et al. (2016), 40%, 25% and 10% of the Amazon basin were affected by moderate, severe and
extreme drought stress in March 2016 when using scPDSI, respectively. This is similar to our estimate (46%, 34% and 9%,
moderately, severely and extremely affected in Sep 2016) based on the same precipitation dataset (ERI). Our estimate
slightly diverted from the results of Jiménez-Muñoz et al. (2016), again at least partly due to a different reference area (see
Methods). In addition, they used spatially resolved information on soil water capacity when calculating scPDSI and a longer
baseline period (start year is 1979 in their study vs. 2000 in our study). scPDSI generally seems to be more sensitive to
baseline changes (Fig S2e). In addition, also the choice of the precipitation dataset plays an important role. In regions, in
which ER5 showed an extremely affected area of only 5%, other datasets such as GLD and TR7 showed a much stronger
drought impact with over 70% of the area moderately and between 50% and 60% severely affected. This is particularly
interesting because recent studies identify TR7, CHR and ER5 as best precipitation datasets when comparing to gauge
observations in South America (Albergel et al., 2018; Burton et al., 2018; Rifai et al., 2019). The higher variability that
scPDSI showed across datasets can be explained with the more complex algorithm (including the self-calibrating
mechanism) compared to MCWD and RAI.

**Implications for estimating drought impacts on the carbon cycle of the Amazon rainforest**

Drought leads to increased tree mortality and carbon losses in tropical forests (Hubau et al., 2020; Lewis et al., 2011; Phillips
et al., 2009). With the prospect of more severe and frequent droughts in a future climate, more precise estimates of how
much carbon is lost from reductions in growth and drought-induced mortality are necessary. Currently, the Amazon
rainforest is acting as a carbon sink, thereby removing $CO_2$ from the atmosphere, but with more frequent and severe drought
events, this sink is already declining (Hubau et al. 2020). Lewis et al. (2011) estimated a total loss of biomass for the
Amazon basin in 2005 of 1.6 Pg C and a 38% more severe impact of 2.2 Pg C for 2010 based on TR6. When applied to the
ΔMCWD derived from the precipitation datasets in our study, we calculate the loss of biomass of the 2005 drought event to
be in the range of 1.3-1.8 Pg C, 1.5-2.3 Pg C in 2010 and 1.3-2.5 Pg C in 2016 (Fig. 4). This corresponds to approximately
the average annual carbon uptake (1-2 PgC) per year, thus, turning the carbon sink into a carbon source. We acknowledge
that our estimates are based on a relatively simple, empirically derived relation that does not take the biomass variability
across the whole Amazon basin and individual forest/tree responses to drought into account. It however gives a rough
estimate of potential carbon losses during drought and an idea of how much it varies depending on the precipitation datasets





applied in a study. In addition, we would like to note that the empirical biomass-MCWD relation from Lewis et al. (2011)
has been estimated with constant ET=100 mm. When using evapotranspiration data (from ER5) for the calculation of
MCWD, we find higher biomass losses (Fig. S2), and thus, the use of MCWD should be carefully viewed via its sensitivity
to ET. In our analysis, MCWD appears to be robust against changes to some parameters, such as baseline period and
inclusion/exclusion of drought years, but to be more sensitive to the evapotranspiration input.

Furthermore, our estimated carbon losses for the drought events might be underestimated as (1) the total duration of the
drought was longer than 12 months (see above paragraph and Fig. 6) and can hence not be fully captured by the standard 12-
month period of the MCWD calculation used in this study, and (2) potential lag effects through delayed plant mortality
within the subsequent years are not considered so far. We would recommend for future studies to investigate the relationship
of biomass losses with other drought indices (such as scPDSI) in a similar manner as done in Lewis et al. (2011). As the
biomass of the Amazon rainforest is heterogeneously distributed (e.g. Saatchi et al., 2011), large-scale biomass-loss induced
by drought (i.e. severe ΔMCWD) should be interpreted carefully. Differences in the amount of biomass in different forest
types, species composition and critical hydraulic processes should be considered when estimating potential biomass losses
under drought stress. A step forward would be to use for example remotely sensed biomass maps to account for regional
biomass distributions (e.g. Avitabile et al., 2016) or to simulate drought impacts with dynamic global vegetation models
(DGVMs). DGVMs simulate the carbon- and water cycle of the biosphere in a process-based way, accounting for the
interplay of carbon uptake and water loss through stomatal opening, evapotranspiration (ET), carbon assimilation via
photosynthesis, and carbon allocation to different plant compartments such as leaves, wood, and roots (e.g. Schaphoff et al.,
2018; Smith et al., 2014). The simulated response of tropical forests in DGVMs is particularly sensitive to precipitation input
under present and future climate change scenarios (e.g. Seiler et al., 2015) and thus, it might be relevant to use multiple
climate forcing datasets to test for climate data uncertainty. Particularly, studies based on ERI and TR6 should possibly be
revisited and include another forcing dataset for their analysis.

## 6. Conclusions

We find substantial variation in the spatial extent, location and timing of the extreme drought events in the years 2005, 2010
and 2016 in the Amazon basin. The variation partly results from the application of different drought metrics (MCWD, RAI,
scPDSI) and from differences in the underlying precipitation datasets. Such differences also propagate when quantifying the
impacts of drought on the carbon cycle of the Amazon rainforest and result in a large variability in biomass carbon losses, as
we show in our analyses. This calls for the application of an ensemble of climate (precipitation) datasets and drought metrics
when assessing the impacts of drought. Communicating the uncertainty in the estimation of drought events and their impacts
on the Amazon rainforest is highly relevant and thus, multiple datasets should be applied by any large-scale study on drought
impacts on vegetation.



**7. Code availability**

All scripts to reproduce analysis and figures are available at https://github.com/PhillipPapastefanou/DroughtAnalysis

**8. Data availability**

All datasets are available following the references in the method section.

**9. Author contribution**

P.P. and A.R. conceived the study and wrote the first draft of the manuscript. All authors contributed to the development of the analysis and the writing of the manuscript.


**10. Competing interests**

The authors declare no competing interests.



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





**Figures**

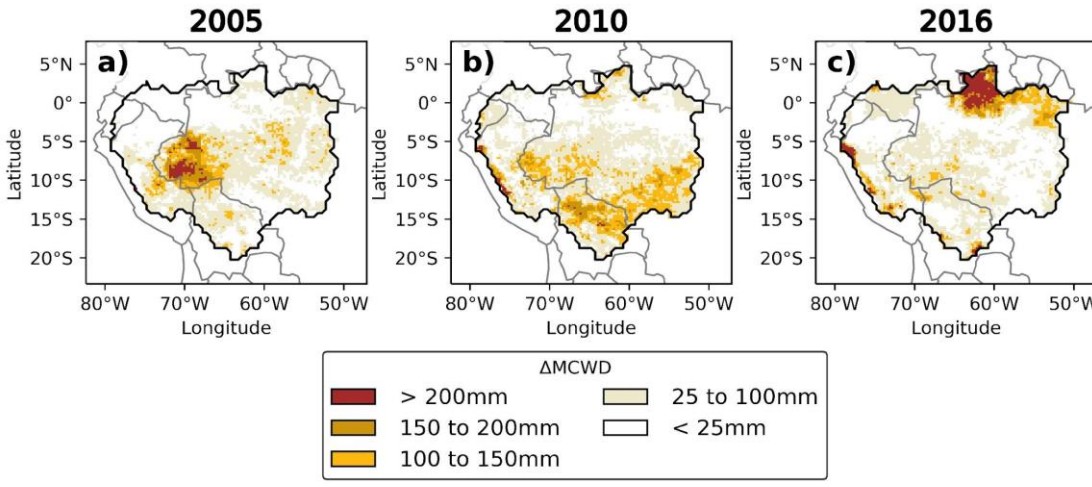


**Figure 1: (a-c)** Anomalies of ΔMCWD (from October to September) as an indicator for drought stress in the Amazon Basin during the record-breaking drought events in 2005, 2010 and 2015/16 based on the TR7 dataset.





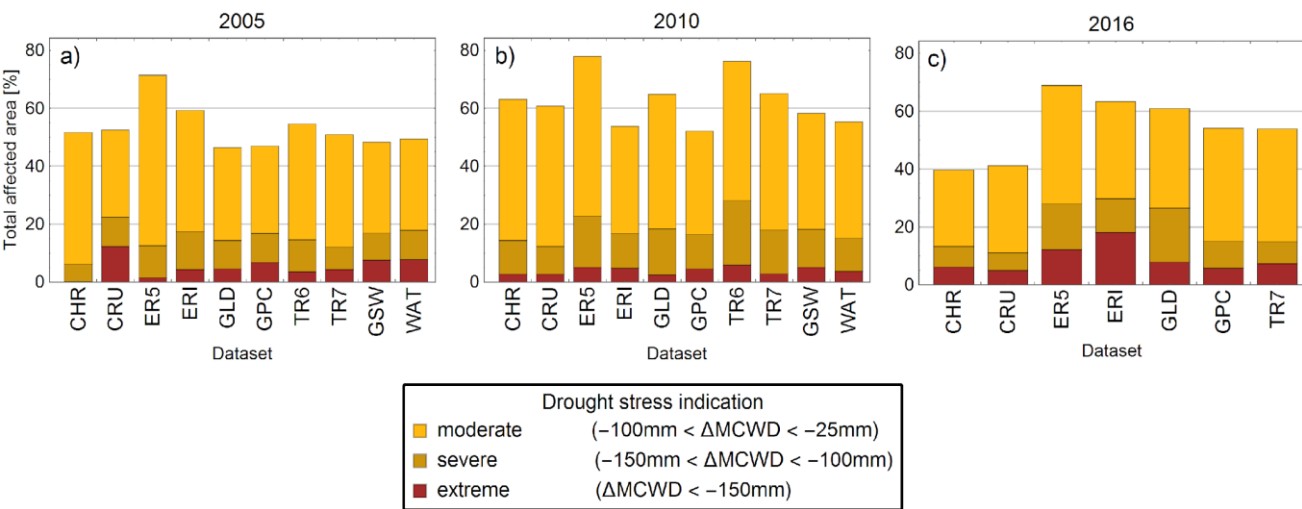


**Figure 2:** Total area of the Amazon basin affected by drought stress (%) according to ΔMCWD for each of the precipitation datasets (for abbreviations see Tab. 1). Displayed are the three drought events (a) 2005, (b) 2010 and (c) 2016. The total area representing the Amazon basin in our study is 5.94 million km². For absolute area affected, see Tab. 2 and 3.
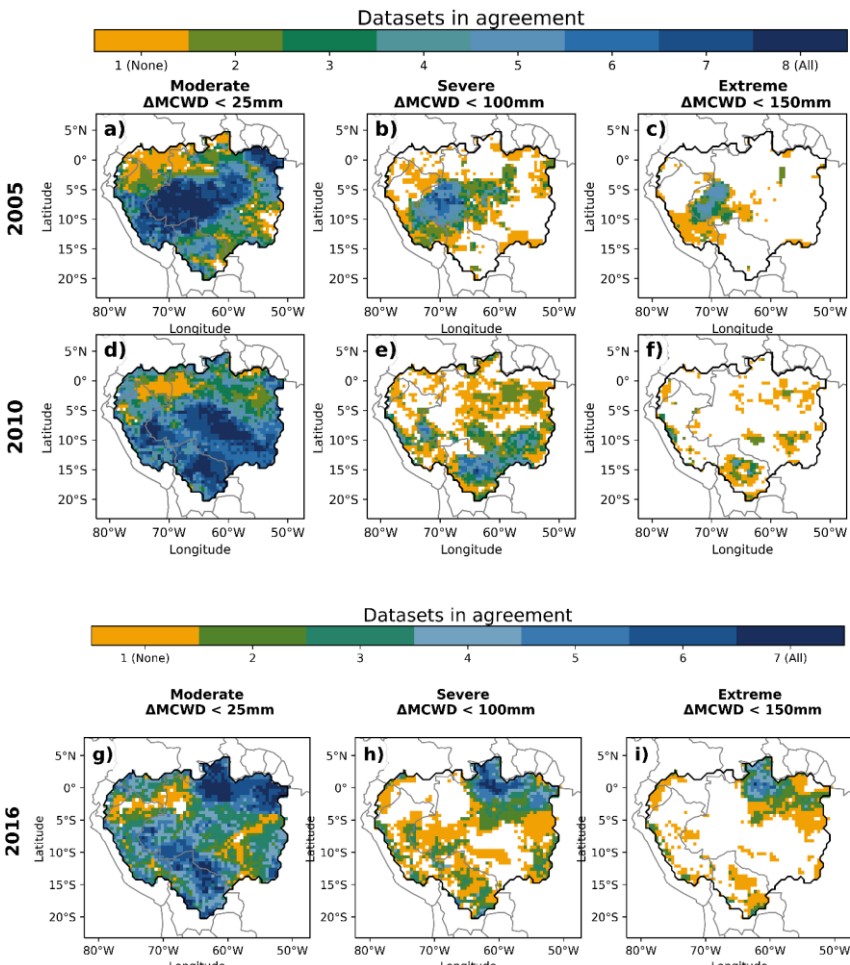


**Figure 3:** Agreement of precipitation datasets on drought area as identified by ΔMCWD anomalies. In columns, different levels of drought severity are displayed and rows show the different drought years 2005 (a-c), 2010 (d-f) and 2016 (g-i). The colors indicate the number of datasets that agree on a specific drought level in a given pixel. Drought severity levels are defined as moderate (ΔMCWD < -25mm), severe (ΔMCWD < -100mm) and extreme (ΔMCWD < -150mm). Orange pixels indicate areas where only dataset shows the respective drought stress (No agreement = "None"). White pixels represent areas where no dataset shows any drought signal. Note that in a-f, TR6 and GSW were excluded, as they were either very similar to its successor (TR7) and or due to a similar reanalysis procedure (WAT). In g-i, only seven datasets were included, which cover the full time period until 2016.





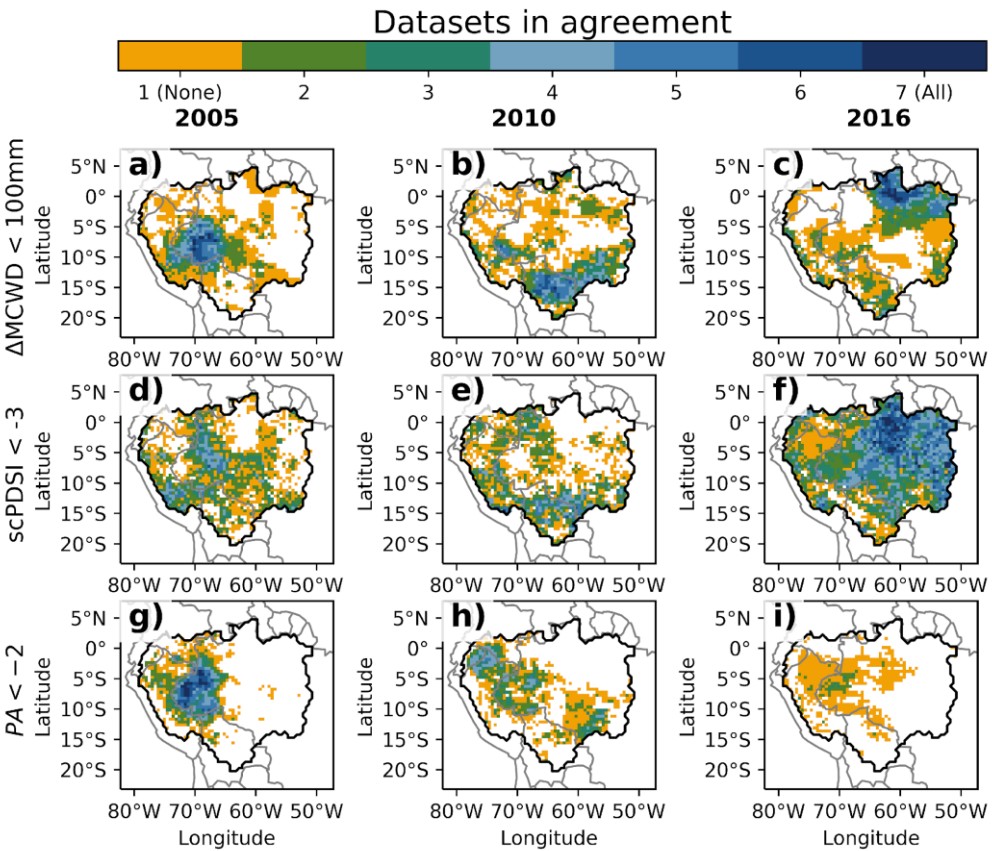

**Figure 4:** Agreement of precipitation datasets on drought area as identified by different drought metrics. Comparison of the
Amazon drought events in 2005, 2010 and 2016 (columns) vs three different drought indexes (rows): ΔMCWD (a-c),
scPDSI (d-f) and rainfall anomaly (g-i). Only the area affected by severe drought stress is displayed, severe drought is
defined differently for each of the drought indices: ΔMCWD less than -100mm, scPDSI less than -3 and RA less than -2.
Orange pixels indicate areas where only one dataset shows the respective drought stress ("None"). White pixels represent
areas where no dataset shows any drought signal.



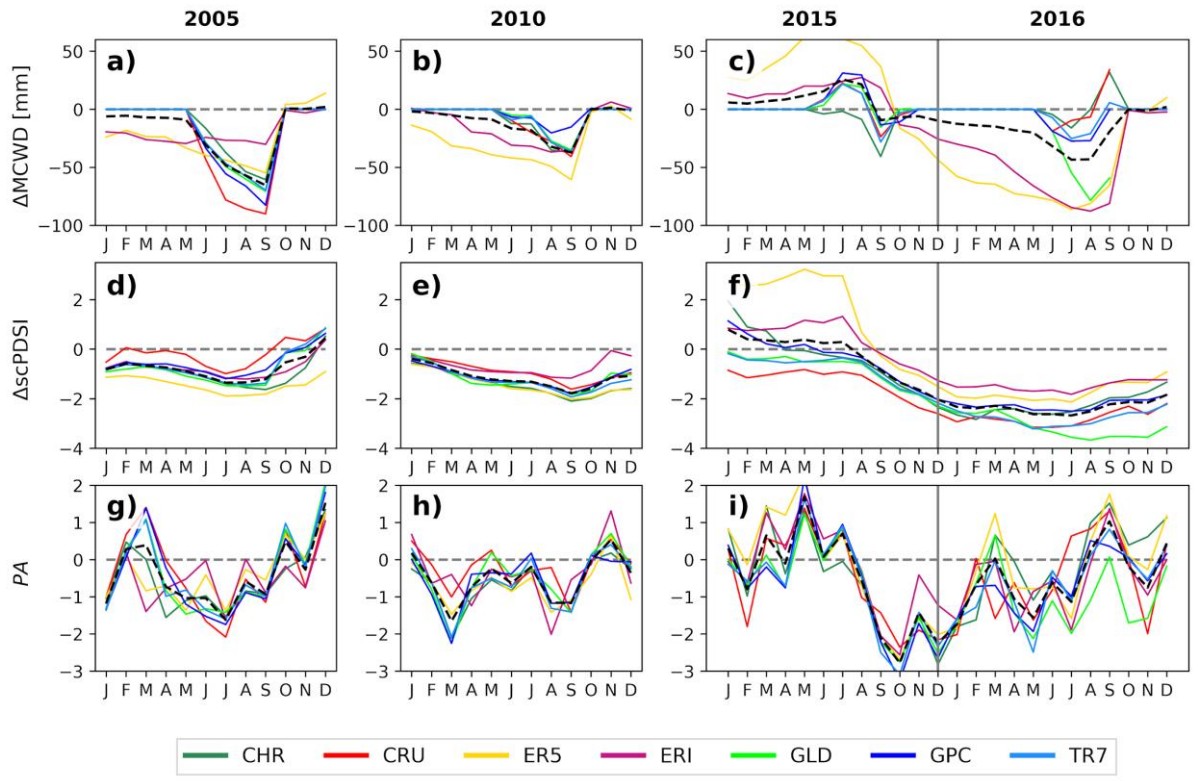

**Figure 5:** Monthly development of the Amazon drought events in 2005, 2010 and 2016 (columns) as described by the three different drought indices (rows): ΔMCWD (a-c), scPDSI (d-f) and rainfall anomaly (RA, g-i). Colored lines indicate the different precipitation datasets (for abbreviations see Tab. 1). RA is estimated for each month.






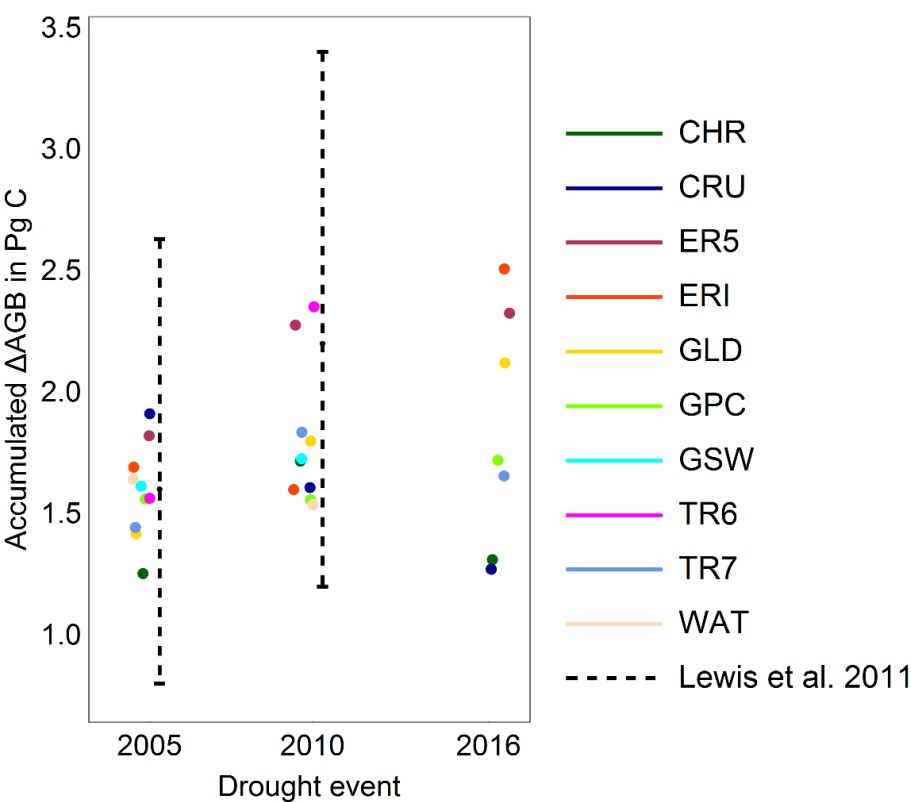

**Figure 6:** Impact of the 2005, 2010 and 2016 drought event on aboveground carbon biomass (AGB in Pg C). Biomass loss was calculated for each of the precipitation datasets (colored dots, for abbreviations see Tab. 1) based on a linear relation between biomass loss and ΔMCWD as proposed by Lewis



**Tables**

**Table 1: Overview of the 10 precipitation datasets used in our study. Columns show the name of the dataset, the official abbreviation, the short abbreviation used in here, the spatial and temporal resolution and the references.**

| Precipitation dataset | Abbreviation | Abbreviation (short) | Details | Resolutions | References |
|---|---|---|---|---|---|
| Climate Hazards group Infrared Precipitation with Stations | CHIRPS | CHR | quasi-global (50°S-50°N) precipitation-only merged product, based on global climatology, satellite estimates and in situ observations. | high resolution (0.05°), daily, pentadal, and monthly | Funk et al., 2015 |
| Tropical Rainfall Measurement Misson | TRMM v6 3b43 | TR6 | quasi-global (50°S-50°N) | Quarter degree resolution (0.25°) daily, pentadal, and monthly | Huffman et al., 2007 |
| Tropical Rainfall Measurement Misson | TRMM v7 3B43 | TR7 | quasi-global (50°S-50°N) | Quarter degree resolution (0.25°), daily, pentadal, and monthly | Huffman et al., 2007 |
| | CRU_NCEP V8 | CNP | global | Half degree resolution (0.5°), daily, pentadal and monthly | Viovy et al., 2017 |
| ERA_Interim | ERA_Interim SFC12_03_TP_228 | ERI | global | 0.75° daily, pentadal, and monthly | Dee et al., 2011 |





| ERA5 | | ER5 | global | Quarter degree resolution (0.25°), sub-daily, daily, monthly | Albergel et al., 2018 |
|---|---|---|---|---|---|
| Global Land Data Assimilation System | GLDAS 2.1 | GLD | global | Quarter degree resolution (0.25°), daily, pentadal, and monthly | Rodell et al., 2004 |
| Global Precipitation Climatology Centre at Deutscher Wetterdienst | GPCC2018 | GPC | global | Quarter degree resolution (0.25°), monthly | Schneider et al., 2018 |
| Global Soil Wetness Project Phase 3 | GSWP3 | GSW | global | Half degree resolution (0.5°), daily, monthly | H. Kim et al. n.d.; http://hydro.iis.u-tokyo.ac.jp/GSWP3/index.html |
| WATCH Forcing Data (WFD) + WATCH Forcing Data methodology applied to ERA-Interim data (WFDEI) | WATCH_WFDEI | WAT | global | Half degree resolution (0.5°), daily, monthly | Weedon et al., 2011, 2014 |





**Table 2: Total area affected by drought stress in million km² (and %) by drought index (MCWD, scPDSI and RAI)**
**and intensity (moderate, severe and extreme) across the 10 datasets evaluated in our study (rows) for the years 2005 and 2010.**

| | | Year | | | | | |
|---|---|---|---|---|---|---|---|
| | | 2005 | 2005 | 2005 | 2010 | 2010 | 2010 |
| Metric | Dataset | $\Delta MCWD <$ $-150mm$ (extreme) | $\Delta MCWD <$ $-100mm$ (severe) | $\Delta MCWD <$ $-25mm$ (moderate) | $\Delta MCWD <$ $-150mm$ (extreme) | $\Delta MCWD <$ $-100mm$ (severe) | $\Delta MCWD <$ $-25mm$ (moderate) |
| ΔMCWD | CHR | 0.0 (0%) | 0.4 (6%) | 3.1 (52%) | 0.2 (3%) | 0.9 (14%) | 3.8 (63%) |
| ΔMCWD | CRU | 0.7 (12%) | 1.3 (22%) | 3.1 (53%) | 0.2 (3%) | 0.7 (12%) | 3.6 (61%) |
| ΔMCWD | ER5 | 0.1 (2%) | 0.7 (13%) | 4.2 (71%) | 0.3 (5%) | 1.3 (23%) | 4.6 (78%) |
| ΔMCWD | ERI | 0.3 (4%) | 1. (17%) | 3.5 (59%) | 0.3 (5%) | 1. (17%) | 3.2 (54%) |
| ΔMCWD | GLD | 0.3 (5%) | 0.9 (14%) | 2.8 (46%) | 0.1 (2%) | 1.1 (18%) | 3.9 (65%) |
| ΔMCWD | GPC | 0.4 (7%) | 1. (17%) | 2.8 (47%) | 0.3 (5%) | 1.0 (16%) | 3.1 (52%) |
| ΔMCWD | TR6 | 0.2 (4%) | 0.9 (15%) | 3.2 (55%) | 0.4 (6%) | 1.7 (28%) | 4.5 (76%) |
| ΔMCWD | TR7 | 0.3 (4%) | 0.7 (12%) | 3. (51%) | 0.2 (3%) | 1.1 (18%) | 3.9 (65%) |
| | | | | | | | |
| ΔMCWD | GSW | 0.5 (8%) | 1.0 (17%) | 2.9 (48%) | 0.3 (5%) | 1.1 (18%) | 3.5 (58%) |
| ΔMCWD | WAT | 0.5 (8%) | 1.1 (18%) | 2.9 (49%) | 0.2 (4%) | 0.9 (15%) | 3.3 (55%) |
| | | $scPDSI <$ $-4$ (extreme) | $scPDSI <$ $-3$ (severe) | $scPDSI <$ $-2$ (moderate) | $scPDSI <$ $-4$ (extreme) | $scPDSI <$ $-3$ (severe) | $scPDSI <$ $-2$ (moderate) |
| scPDSI | CHR | 0.2 (3%) | 1.2 (20%) | 2.5 (42%) | 0.2 (3%) | 2. (34%) | 3.2 (55%) |
| scPDSI | CRU | 0.3 (4%) | 1.5 (26%) | 2.3 (38%) | 0.1 (2%) | 2. (33%) | 3.1 (52%) |
| scPDSI | ER5 | 0.1 (1%) | 1.1 (18%) | 2.8 (46%) | 0.1 (1%) | 1.6 (27%) | 3.1 (52%) |
| scPDSI | ERI | 0.0 (1%) | 0.8 (13%) | 1.7 (29%) | 0. 0(1%) | 1.2 (20%) | 2.1 (35%) |
| scPDSI | GLD | 0.2 (3%) | 1.0 (16%) | 1.9 (32%) | 0.2 (3%) | 2.9 (50%) | 4.2 (71%) |
| scPDSI | GPC | 0.1 (2%) | 1.5 (25%) | 2.6 (43%) | 0.1 (3%) | 1.9 (32%) | 3. (51%) |
| scPDSI | TR6 | 0.3 (5%) | 1.5 (25%) | 2.8 (48%) | 0.2 (3%) | 1.9 (32%) | 3.2 (54%) |
| scPDSI | TR7 | 0.3 (5%) | 1.5 (25%) | 2.8 (48%) | 0.2 (3%) | 1.9 (32%) | 3.2 (54%) |
| scPDSI | GSW | 0.2 (3%) | 1.6 (26%) | 2.6 (44%) | 0.2 (3%) | 1.8 (31%) | 3.1 (52%) |
| scPDSI | WAT | 0.2 (3%) | 1.5 (26%) | 2.6 (44%) | 0.2 (3%) | 1.8 (30%) | 3. (51%) |





| | | $RA < -3$ (extreme) | $RA < -2$ (severe) | $RA < -1$ (moderate) | $RA < -3$ (extreme) | $RA < -2$ (severe) | $RA < -1$ (moderate) |
|---|---|---|---|---|---|---|---|
| RA | CHR | 0.3 (6%) | 1.2 (20%) | 3.1 (52%) | 0.2 (3%) | 1. (17%) | 3.6 (60%) |
| RA | CRU | 0.1 (2%) | 0.6 (9%) | 1.8 (29%) | 0.1 (1%) | 1. (17%) | 3. (50%) |
| RA | ER5 | 0.3 (4%) | 1.1 (18%) | 2.9 (49%) | 0.4 (6%) | 1.7 (28%) | 4.2 (71%) |
| RA | ERI | 0.6 (10%) | 1.1 (18%) | 2.5 (42%) | 0.2 (3%) | 1.0 (16%) | 2.7 (45%) |
| RA | GLD | 0.2 (4%) | 0.7 (12%) | 1.7 (29%) | 0.6 (9%) | 1.2 (21%) | 3.4 (57%) |
| RA | GPC | 0.2 (4%) | 0.7 (11%) | 2.2 (36%) | 0.1 (2%) | 0.7 (12%) | 2.7 (46%) |
| RA | TR6 | 0.1 (2%) | 0.6 (11%) | 2.4 (41%) | 0.1 (2%) | 1.3 (22%) | 3.7 (63%) |
| RA | TR7 | 0.2 (3%) | 0.9 (15%) | 2.8 (47%) | 0.2 (4%) | 1.2 (20%) | 3.3 (56%) |
| RA | GSW | 0.2 (4%) | 0.7 (11%) | 2.1 (36%) | 0.2 (3%) | 0.9 (16%) | 3.1 (52%) |
| RA | WAT | 0.3 (4%) | 0.7 (12%) | 2.2 (37%) | 0.1 (2%) | 0.8 (13%) | 2.8 (47%) |

**Table 3: Total area affected by drought in million km² (and %) by drought index (MCWD, scPDSI and RAI) and intensity (moderate, severe and extreme) across the 10 datasets evaluated in this study (rows) for the year 2016. TR6, GSW and WAT are missing from this calculation as their timespan ends before 2016.**

| | | 2016 | 2016 | 2016 |
|---|---|---|---|---|
| Metric | Dataset | $\Delta MCWD$ $< -150mm$ (extreme) | $\Delta MCWD$ $< -100mm$ (severe) | $\Delta MCWD$ $< -25mm$ (moderate) |
| ΔMCWD | CHR | 0.4 (6%) | 0.8 (13%) | 2.4 (40%) |
| ΔMCWD | CRU | 0.3 (5%) | 0.7 (11%) | 2.4 (41%) |
| ΔMCWD | ER5 | 0.7 (12%) | 1.7 (28%) | 4.1 (69%) |
| ΔMCWD | ERI | 1.1 (18%) | 1.8 (30%) | 3.8 (63%) |
| ΔMCWD | GLD | 0.5 (8%) | 1.6 (27%) | 3.6 (61%) |
| ΔMCWD | GPC | 0.3 (6%) | 0.9 (15%) | 3.2 (54%) |
| ΔMCWD | TR7 | 0.4 (7%) | 0.9 (15%) | 3.2 (54%) |
| | | $scPDSI < -4$ (extreme) | $scPDSI < -3$ (severe) | $scPDSI < -2$ (moderate) |
| scPDSI | CHR | 0.3 (4%) | 2.3 (38%) | 3.3 (56%) |
| scPDSI | CRU | 0.3 (5%) | 2.6 (45%) | 3.7 (62%) |





| scPDSI | ER5 | 0.3 (5%) | 2.1 (35%) | 2.9 (48%) |
|--------|-----|----------|-----------|-----------|
| scPDSI | ERI | 0.5 (9%) | 2. (34%) | 2.7 (46%) |
| scPDSI | GLD | 0.9 (15%) | 3.7 (62%) | 4.2 (70%) |
| scPDSI | GPC | 0.4 (7%) | 2.3 (39%) | 3.2 (55%) |
| scPDSI | TR7 | 0.6 (11%) | 3.1 (52%) | 4.2 (71%) |
| | | $RA < -3$ (extreme) | $RA < -2$ (severe) | $RA < -1$ (moderate) |
| RA | CHR | 0.0 (0%) | 0.1 (2%) | 0.5 (8%) |
| RA | CRU | 0.0 (0%) | 0.0 (0%) | 0.3 (5%) |
| RA | ER5 | 0.0 (0%) | 0.0 (0%) | 0.5 (9%) |
| RA | ERI | 0.0 (0%) | 0.0 (1%) | 0.9 (15%) |
| RA | GLD | 0.6 (10%) | 1.8 (30%) | 3.2 (54%) |
| RA | GPC | 0.0 (0%) | 0.0 (0%) | 0.7 (12%) |
| RA | TR7 | 0.0 (0%) | 0.1 (1%) | 0.9 (14%) |
