# Peer review of "Recent extreme drought events in the Amazon rainforest: Assessment of"

_Biogeosciences, 2020_

## Referee Comment (RC1) · Xiangtao Xu (Referee) · 11 Mar 2021

Xiangtao Xu (Referee)

xu.withoutwax@gmail.com

*General comments*:

Papastefanou et al. assessed the extent and severity of the 2005,2010, and 2015/2016 droughts over the Amazon basin using 10 precipitation data sources and 3 drought indexes (MCWD, scPDSI, and RAI) with different assumptions. The main results show an increasing disagreement across datasets for more severe drought signals (in terms of both frequency and location). PDSI which consider variable ET shows a much stronger drought impact in 2016 compared with MCWD while RAI based on dry season rainfall shows a weaker drought impact in 2016. In addition, the research explored the consequences of estimating biomass loss from uncertainty across different precipitation using an empirical drought-mortality relationship. The resultant uncertainty in total carbon loss can reach 1.4 PgC (1.3-2.7) for the 2015/2016 drought. The authors conclude with a recommendation of using an ensemble of precipitation data sets when assessing the impact of drought.

Overall, I think the analysis is a useful contribution to the study of drought impacts over the Amazon or more generally the tropical forests. The research provides a comprehensive overview of the differences across rainfall datasets, an issue that any analysis or modeling studies over tropical drought will struggle with.

I feel the key figures showing dataset agreement are helpful. However, I think the manuscript can benefit from more in-depth discussion and a stronger conclusion. Please see the below specific comments for details. Hopefully, they will help to improve the manuscript and make it more useful to the scientific community.

*Specific Comments*:

1. The manuscript focuses on the disagreement among drought indices across different precipitation data sets, which are ultimately driven by the differences in precipitation. It would be helpful to show the difference (e.g. systematic biases and spatial-temporal correlation) across the raw precipitation data sets using paired scatter plots for each precipitation data combination (could be put in the supplementary). This can help to understand why there are disagreements in MCWD (is it just because of a systematic bias so certain data set generates lower MCWD or due to disagreement in the spatial distribution of rainfall, etc.)Such analyses can help to illustrate.

A related point is how to compare different drought indices. Current categorization into moderate, severe, and extreme seems too subjective. Why not show the scatter plot between different drought indices across the drought (from selected precipitation dataset or averaged across all precipitation datasets), which can show the scaling between MCWD, scPDSI, and RAI and demonstrates their differences. Or maybe use

percentile (e.g. lowest 5% to indicate extreme) to compare across indices?

2. I like the idea of translating uncertainty in MCWD into the uncertainty in AGB changes (ln 215). However, it should be acknowledged that the empirical relationship itself subjects to large uncertainty. For example, Feldpausch et al. (2016) find that the mortality-MCWD relationship identified in 2005 disappeared during the 2010 drought.

Feldpausch T R, Phillips O L, Brienen R J W, Gloor E, Lloyd J, Lopez-Gonzalez G, Monteagudo-Mendoza A, Malhi Y, Alarcón A, Álvarez Dávila E, Alvarez-Loayza P, Andrade A, Aragao L E O C, Arroyo L, Aymard C. G A, Baker T R, Baraloto C, Barroso J, Bonal D, Castro W, Chama V, Chave J, Domingues T F, Fauset S, Groot N, Honorio Coronado E, Laurance S, Laurance W F, Lewis S L, Licona J C, Marimon B S, Marimon-Junior B H, Mendoza Bautista C, Neill D A, Oliveira E A, Oliveira dos Santos C, Pallqui Camacho N C, Pardo-Molina G, Prieto A, Quesada C A, Ramírez F, Ramírez-Angulo H, Réjou-Méchain M, Rudas A, Saiz G, Salomão R P, Silva-Espejo J E, Silveira M, ter Steege H, Stropp J, Terborgh J, Thomas-Caesar R, van der Heijden G M F, Vásquez Martinez R, Vilanova E and Vos V A 2016 Amazon forest response to repeated droughts Global Biogeochem. Cycles 30 964–82 Online: https://agupubs.onlinelibrary.wiley.com/doi/full/10.1002/2015GB005133

In addition, I am not sure whether directly plugging in MCWD based on different rainfall data set makes sense. eqn 2 was derived using a specific rainfall data set. I think it would make more sense to remove the systematic biases between the specific data set and all the data set used in this study before converting MCWD to AGB. One way to find the mapping between MCWD data sets is simple regressions between the data sets as suggested in my comment above. Will such cross-data set calibration reduce AGB uncertainty?

3. Current conclusion recommends using an ensemble of different rainfall data sets when analyzing drought impacts. However, is there strong evidence that the ensemble would perform better than individual data sets? I wonder whether there are ways to

evaluate the performance of each rainfall data set in terms of estimating drought impact. For example, is it possible to compare the spatial and temporal patterns of AGB loss based on different rainfall data sets with the observed spatial-temporal patterns from microwave remote sensing data (Liu et al. 2015; Saatchi et al. 2013; Wigneron et al. 2020) or lidar data (Yang et al. 2018)? Some more detailed details on the potential biases of MCWD that do not include ET variability?

Liu Y Y, Van Dijk A I J M, De Jeu R A M, Canadell J G, McCabe M F, Evans J P and Wang G 2015 Recent reversal in loss of global terrestrial biomass Nat. Clim. Chang. 5 470–4 Online: http://dx.doi.org/10.1038/nclimate2581

Saatchi S, Asefi-Najafabady S, Malhi Y, Aragão L E O C, Anderson L O, Myneni R B and Nemani R 2013 Persistent effects of a severe drought on Amazonian forest canopy Proc. Natl. Acad. Sci. U. S. A. 110 565–70 Online: http://www.ncbi.nlm.nih.gov/pubmed/23267086

Wigneron J P, Fan L, Ciais P, Bastos A, Brandt M, Chave J, Saatchi S, Baccini A and Fensholt R 2020 Tropical forests did not recover from the strong 2015–2016 El Niño event Sci. Adv. 6 eaay4603 Online: https://advances.sciencemag.org/content/6/6/eaay4603

Yang Y, Saatchi S S, Xu L, Yu Y, Choi S, Phillips N, Kennedy R, Keller M, Knyazikhin Y and Myneni R B 2018 Post-drought decline of the Amazon carbon sink Nat. Commun. 9 3172 Online: http://www.nature.com/articles/s41467-018-05668-6

4. ln 369, I thought microwave data is mostly free from cloud cover effect, which mainly influence optical remote sensing products? I think some of the challenges are the limited penetration depth in the dense tropical forests (Chaparro et al. 2019) and the influences of vegetation water status (Xu et al. 2021)

Chaparro D, Duveiller G, Piles M, Cescatti A, Vall-llossera M, Camps A and Entekhabi D 2019 Sensitivity of L-band vegetation optical depth to carbon stocks in tropical

forests: a comparison to higher frequencies and optical indices Remote Sens. Environ. 232 111303

Xu X, Konings A G, Longo M, Feldman A, Xu L, Saatchi S, Wu D, Wu J and Moorcroft P 2021 Leaf surface water, not plant water stress, drives diurnal variation in tropical forest canopy water content New Phytol.

5. ln 424-425, as I argued in my second comment, I am not sure whether it makes sense to directly apply MCWD based on variable ET onto a relationship based on MCWD based on fixed ET....

*Stylistic Comments and Technical Corrections:*

ln 63: 'altering the carbon cycle of the Amazon forest already today' -> 'already altering the carbon cycle of the Amazon forest'

ln 80-100: I wonder whether it is better to just briefly talk about the usage of ten different data sets here and move the details into Methods

ln 122: 0.6 Mio -> 0.6 million?

ln 402: 'In addition, also', the also is extra

ln 419: 'average annual carbon uptake' global or regional? Please specify

I wonder whether Table 2 and Table 3 are more suitable for SI... Especially if additional figures on the difference across rainfall datasets are added in the revision.

―――――――――――――――――――

---

## Referee Comment (RC2) · Anonymous Referee #2 · 26 May 2021

The authors present a comparison of drought metrics, calculated with different rainfall products. The study region is focused on the Amazon basin, and an extrapolation is made of aboveground forest carbon loss from drought. The authors end with a message that evaluation of drought through an ensemble is better.

I think the comparison of rainfall products and evaluation of drought metrics could be useful, especially if it is more developed in the revision. This section could use some more analysis, especially with respect to defining anomalies per pixel location rather than absolute thresholds. However the section concerning the extrapolation of forest carbon loss from drought is a large overreach and does not help advance the state of

the science. Please see the following general comments, and line comments.

General comments: Carbon loss from drought - I will start with my strongest objection to this study, which is the extrapolation of forest carbon loss from drought. Accurate estimation of tropical forest carbon loss from drought is a highly sought after goal for tropical ecosystem ecology, but the methods this study uses are not robust or defensible in the present day. The standing biomass and forest sensitivity to drought differs dramatically across Amazonia. This point is even acknowledged (Line 435) in the manuscript. This study does not present any new field data to evaluate this very simplistic empirical relationship (from Lewis 2011), and therefore this study does not have the substance to make these claims. Even Lewis (2011) states this is a first approximation approach and does not include any goodness of fit statistics, the number of plots used to derive this estimate, or even specific information about which RAINFOR plots were included. Lewis extrapolated the relationship beyond the MCWD observed within the RAINFOR plot network from the 2005 drought through the 2010 drought to produce a quick estimate of carbon loss. In this study, the simplistic linear relationship is extrapolated even further beyond the original Lewis 2011 extrapolation. Even if this original relationship was remotely accurate for the 2005 drought, there is no evidence that it was accurate for subsequent droughts in 2010 (or 2015/16). It is difficult to make these forest carbon loss estimates regarding the 2015/16 drought without new field observations and validation, therefore I do not agree that the AGB loss estimates presented here are justifiable and object to their inclusion.

Next, it is worth noting that a large-scale squall line also crossed the Amazon basin during the period of measurements presented in the original Phillips 2009 Science paper. This was estimated to have killed hundreds of millions of trees (Negrón-Juárez et al., 2010 Geophysical Research Letters), so even the empirical AGB∼MCWD loss relationship presented in Lewis 2011 has a heavy bias from wind mortality. I strongly urge the authors to drop this aspect of the manuscript. Estimating Amazonian forest carbon loss from drought has long been a difficult endeavour, and many groups have

been physically collecting field observations to quantify this. I worry this aspect of the study adds more noise than value to the current state of the science.

Defining drought - I think the evaluation of different precipitation datasets concerning the drought is mostly fine and could be useful. However the way drought is defined here is a bit simplistic, especially regarding the MCWD anomaly. The mean annual precip spans from 3500 mm + in the northwest Amazon to less than 1700 mm in the southeastern peripheries. I think it is difficult to justify a definition of drought based on absolute thresholds for the MCWD anomaly. The northwest Amazon rarely experiences a dry season, whereas the southeast Amazon does not receive rainfall for more than half the year. Forests are adapted to some level of water stress, which is why simple absolute thresholds are unlikely to characterize vegetation water stress. Assessing drought anomalies based on the number of standard deviations (calculated per pixel-location) is one commonly used way to assess drought with respect to the baseline climate and interannual variability of precipitation.

Absolute thresholds (e.g. MCWD >25) vs. relative anomalies (e.g. MCWD > 2 standard deviations). The older papers using MCWD (e.g. Aragão et al., 2007) used a fixed value because there was not enough information at the time of actual ET. Now it is well understood that actual ET can vary substantially across the Amazon and has seasonality in most regions. It no longer makes sense to use a fixed value of ET for both the everwet northwest Amazon and the seasonally dry southwest Amazon. I suggest the authors could use newer spatially resolved ET estimates such as from GLEAM, MODIS MOD16, Fluxcom, etc.

The comparison of precipitation products and drought metrics could be a useful contribution, however this is currently muddled by putting all the estimates together in an ensemble. I suggest the authors focus on presenting a more organized comparison of (1) precipitation products, and (2) drought metrics. What is the justification for using an ensemble of precipitation datasets? Why is this better than using the best evaluated precipitation dataset? Consider the timing of the development of these products. Some

of them have been operational for over 20 years. Statistical methods, data assimilation and climate reanalysis models have improved dramatically since then. I think it is difficult to argue that an ensemble method is better, especially when including where a coarse resolution earlier generation product (e.g. GPCC) has as much vote as the latest generation of products (e.g. ERA5, GPM IMERG6).

Other comments - There are a number of typos in both the main text and figures. Some of these are highlighted in the line comments.

There are far too many acronyms in this manuscript. For example, is CHR really a useful shortening of the CHIRPS? Each new acronym makes the manuscript more difficult to read. I suggest limiting the usage of acronyms to the absolute minimum. Wherever possible, use established acronyms such as TRMM. Making up new acronyms of acronyms (TR6, TR7) is confusing and will not help readers comprehend the manuscript. A manuscript of this length does not need additional acronyms to make it shorter.

Section comments: L30: This should be MCWD > 25 mm, no? Also the climatological mean MCWD across Amazonia is quite large. I don't think it makes sense to use a single value to define drought (∼25 mm). MCWD >= 25 mm in the southeast Amazon does not indicate drought.

L 170: The wet season starts at different times of the year across the Amazon. How is the choice of starting the hydrological year determined?

L 173: I am not sure Delta MCWD is a good abbreviation for the anomaly of MCWD. This can easily be taken as just the change in MCWD between two time periods, but that's not exactly what the anomaly is during a drought. Perhaps it's better to spell it out as the " MCWD anomaly".

L 176: Removing the drought years causes bias. There are three droughts in the span of 15 years, so these are not rare events. Just because Lewis 2011 used a method,

does not mean it is defensible in the present day.

L 185: Climatologies are typically calculated from 30 year periods. Most of the data products have at least 20 years of duration, if not closer to 40. The selection of years to remove is subjective and removing the years with anomalously low rainfall will bias the standard deviation to be artificially small.

L208: Be consistent in treating MCWD as either a positive or negative quantity.

L215+: I reject the underlying basis for the empirical carbon loss estimate from Lewis (2011).

L229: MCWD is misspelled

L295: It is difficult for rainfall products to correctly estimate rainfall near the foothills of the Andes. Also some areas have very little ground information for each product's bias correction algorithm. It might be worth getting into this to describe more deeply why the products disagree, and where.

L333: I would note that many studies no longer use the fixed estimate of 100 mm. I believe some have used Stephenson (1998 Journal of Biogeography) as a reference for the development of the MCWD metric.

L357: Using a better estimate of "actual ET" might reflect the impact of VPD. I would say this is a limitation of using a fixed 100mm value for ET in the MCWD calculation.

L426: Indeed, this is another reason to drop the extrapolated carbon loss estimates.

L453: I don't think the case for assessing drought with an ensemble is made clear. Why is it not better to just use the product that has the lowest RMSE in the region of interest?

L458: The code in the repo looks to be incomplete. Ideally the complete code for analysis and figures should be hosted prior to the review process. An incomplete repository hinders the review process.

Figure 1: Is this MCWD, or anomalies of MCWD?

Figure 2: Why is WAT not included in panel C?

Figure 3: This is a useful figure. It might be useful to add another two columns indicating where the satellite based products agree, and where the climate reanalysis modeled products agree.

Figure 4: Is "PA" (y-axis label) supposed to be "RAI"?

Figure 5: Is "PA" (y-axis label) supposed to be "RAI"? The delta MCWD supposed to be the Anomaly of MCWD? Might be better to spell this out.

Figure 6: I suggest removing this aspect of the study, and this figure.

Table 1: I suggest dropping the abbreviations of abbreviations, and adding a column about how the product is derived (e.g. Remote sensing, interpolation of ground data, atmospheric process model, etc).

Table 2: RAI?

---

## Author Comment (AC1) · 24 Jun 2021

**General comments*:**

Papastefanou et al. assessed the extent and severity of the 2005,2010, and 2015/2016 droughts over the Amazon basin using 10 precipitation data sources and 3 drought indexes (MCWD, scPDSI, and RAI) with different assumptions. The main results show an increasing disagreement across datasets for more severe drought signals (in terms of both frequency and location). PDSI which consider variable ET shows a much stronger drought impact in 2016 compared with MCWD while RAI based on dry season rainfall shows a weaker drought impact in 2016. In addition, the research explored the consequences of estimating biomass loss from uncertainty across different precipitation using an empirical drought-mortality relationship. The resultant uncertainty in total carbon loss can reach 1.4 PgC (1.3-2.7) for the 2015/2016 drought. The authors conclude with a recommendation of using an ensemble of precipitation data sets when assessing the impact of drought. Overall, I think the analysis is a useful contribution to the study of drought impacts over the Amazon or more generally the tropical forests. The research provides a comprehensive overview of the differences across rainfall datasets, an issue that any analysis or modeling studies over tropical drought will struggle with. I feel the key figures showing dataset agreement are helpful. However, I think the manuscript can benefit from more in-depth discussion and a stronger conclusion. Please see the below specific comments for details. Hopefully, they will help to improve the manuscript and make it more useful to the scientific community.

- *We thank the reviewer for his constructive feedback. We address all comments in detail in the sections below.*

**Specific Comments*:**

1. The manuscript focuses on the disagreement among drought indices across different precipitation data sets, which are ultimately driven by the differences in precipitation. It would be helpful to show the difference (e.g. systematic biases and spatialtemporal correlation) across the raw precipitation data sets using paired scatter plots for each precipitation data combination (could be put in the supplementary). This can help to understand why there are disagreements in MCWD (is it just because of a systematic bias so certain data set generates lower MCWD or due to disagreement in the spatial distribution of rainfall, etc.) Such analyses can help to illustrate.

- *We agree with the reviewer that analysing the precipitation datasets in more detail will improve the understanding of the differences of MCWD. We will create additional scatterplots showing the differences across precipitation datasets to identify potential biases. The plots will be added to the supplementary material.*

A related point is how to compare different drought indices. Current categorization into moderate, severe, and extreme seems too subjective. Why not show the scatter plot between different drought indices across the drought (from selected precipitation dataset or averaged across all precipitation datasets), which can show the scaling between MCWD, scPDSI, and RAI and demonstrates their differences. Or maybe use percentile (e.g. lowest 5% to indicate extreme) to compare across indices?

- *We agree with the reviewer that our categorization is subjective. This was also pointed out by reviewer 2. We thus will change our analysis and use relative instead of absolute thresholds to enable better cross-comparison of the drought indices.*

2. I like the idea of translating uncertainty in MCWD into the uncertainty in AGB changes (ln 215). However, it should be acknowledged that the empirical relationship itself subjects to large uncertainty. For example, Feldpausch et al. (2016) find that the mortality-MCWD relationship identified in 2005 disappeared during the 2010 drought. Feldpausch T R, Phillips O L, Brienen R J W, Gloor E, Lloyd J, Lopez-Gonzalez G, Monteagudo-Mendoza A, Malhi Y, Alarcón A, Álvarez Dávila E, Alvarez-Loayza P, Andrade A, Aragao L E O C, Arroyo L, Aymard C. G A, Baker T R, Baraloto C, Barroso J, Bonal D, Castro W, Chama V, Chave J, Domingues T F, Fauset S, Groot N, Honorio Coronado E, Laurance S, Laurance W F, Lewis S L, Licona J C, Marimon B S, Marimon-Junior B H, Mendoza Bautista C, Neill D A, Oliveira E A, Oliveira dos Santos C, Pallqui Camacho N C, Pardo-Molina G, Prieto A, Quesada C A, Ramírez F, Ramírez-Angulo H, Réjou-Méchain M, Rudas A, Saiz G, Salomão R P, Silva-Espejo J E, Silveira M, ter Steege H, Stropp J, Terborgh J, Thomas-Caesar R, van der Heijden G M F, Vásquez Martinez R, Vilanova E and Vos V A 2016 Amazon forest response to repeated droughts Global Biogeochem. Cycles 30 964–82 Online: https://agupubs.onlinelibrary.wiley.com/doi/full/10.1002/2015GB005133 In addition, I am not sure whether directly plugging in MCWD based on different rainfall data set makes sense. eqn 2 was derived using a specific rainfall data set. I think it would make more sense to remove the systematic biases between the specific data set and all the data set used in this study before converting MCWD to AGB. One way to find the mapping between MCWD data sets is simple regressions between the data sets as suggested in my comment above. Will such cross-data set calibration reduce AGB uncertainty?

- *We thank the author for highlighting the Feldpausch et al. 2016 study which we missed when writing our manuscript and we agree that the linear relation between AGB and MCWD does not hold for 2010 and 2016. This was also pointed out by referee 2. We thus will remove the AGB loss estimates for 2010 and 2016.*
- *We appreciate the suggestion of the referee to deeper investigate the MCWD-AGB relation using multiple precipitation datasets and we would be happy to work on this together in a follow-up study.*

3. Current conclusion recommends using an ensemble of different rainfall data sets when analyzing drought impacts. However, is there strong evidence that the ensemble would perform better than individual data sets? I wonder whether there are ways to evaluate the performance of each rainfall data set in terms of estimating drought impact. For example, is it possible to compare the spatial and temporal patterns of AGB loss based on different rainfall data sets with the observed spatial-temporal patterns from microwave remote sensing data (Liu et al. 2015; Saatchi et al. 2013; Wigneron et al. 2020) or lidar data (Yang et al. 2018)? Some more detailed details on the potential biases of MCWD that do not include ET variability?

- *We thank the referee for these important remarks. While we do not want to state that an ensemble (collection of datasets) generally performs better than one single dataset, our point is that drought stress can differ substantially between datasets. So for studies assessing impacts of droughts on the Amazon rainforest it may be worth considering multiple datasets to test for climate uncertainty purely arising by the choice of precipitation dataset. We will reformulate our manuscript accordingly.*
- *We appreciate the reviewers idea regarding the comparisons to remote sensing data. While this would probably go beyond the scope of this study we think that it would be interesting to investigate in a follow-up study.*

**Referees*:**

Liu Y Y, Van Dijk A I J M, De Jeu R A M, Canadell J G, McCabe M F, Evans J P and Wang G 2015 Recent reversal in loss of global terrestrial biomass Nat. Clim. Chang. 5 470–4 Online: http://dx.doi.org/10.1038/nclimate2581

Saatchi S, Asefi-Najafabady S, Malhi Y, Aragão L E O C, Anderson L O, Myneni R B and Nemani R 2013 Persistent effects of a severe drought on Amazonian forest canopy Proc. Natl. Acad. Sci. U. S. A. 110 565–70 Online: http://www.ncbi.nlm.nih.gov/pubmed/23267086

Wigneron J P, Fan L, Ciais P, Bastos A, Brandt M, Chave J, Saatchi S, Baccini A and Fensholt R 2020 Tropical forests did not recover from the strong 2015–2016 El Niño event Sci. Adv. 6 eaay4603 Online: https://advances.sciencemag.org/content/6/6/eaay4603

Yang Y, Saatchi S S, Xu L, Yu Y, Choi S, Phillips N, Kennedy R, Keller M, Knyazikhin Y and Myneni R B 2018 Post-drought decline of the Amazon carbon sink Nat. Commun. 9 3172 Online: http://www.nature.com/articles/s41467-018-05668-6 4.

In 369, I thought microwave data is mostly free from cloud cover effect, which mainly influence optical remote sensing products? I think some of the challenges are the limited penetration depth in the dense tropical forests (Chaparro et al. 2019) and the influences of vegetation water status (Xu et al. 2021)

Chaparro D, Duveiller G, Piles M, Cescatti A, Vall-llossera M, Camps A and Entekhabi D 2019 Sensitivity of L-band vegetation optical depth to carbon stocks in tropical forests: a comparison to higher frequencies and optical indices Remote Sens. Environ. 232 111303

Xu X, Konings A G, Longo M, Feldman A, Xu L, Saatchi S, Wu D, Wu J and Moorcroft P 2021 Leaf surface water, not plant water stress, drives diurnal variation in tropical forest canopy water content New Phytol. 5. ln 424-425, as I argued in my second comment, I am not sure whether it makes sense to directly apply MCWD based on variable ET onto a relationship based on MCWD based on fixed ET....

**Stylistic Comments and Technical Corrections:**

ln 63: 'altering the carbon cycle of the Amazon forest already today' -> 'already altering the carbon cycle of the Amazon forest'
ln 80-100: I wonder whether it is better to just briefly talk about the usage of ten different data sets here and move the details into Methods
ln 122: 0.6 Mio -> 0.6 million?
ln 402: 'In addition, also', the also is extra
ln 419: 'average annual carbon uptake' global or regional? Please specify I wonder whether Table 2 and Table 3 are more suitable for SI... Especially if additional figures on the difference across rainfall datasets are added in the revision.

- *We thank the reviewer for the detailed comments and corrections which we will fix accordingly.*

---

## Author Comment (AC2) · 24 Jun 2021

The authors present a comparison of drought metrics, calculated with different rainfall products. The study region is focused on the Amazon basin, and an extrapolation is made of aboveground forest carbon loss from drought. The authors end with a message that evaluation of drought through an ensemble is better. I think the comparison of rainfall products and evaluation of drought metrics could be useful, especially if it is more developed in the revision. This section could use some more analysis, especially with respect to defining anomalies per pixel location rather than absolute thresholds. However the section concerning the extrapolation of forest carbon loss from drought is a large overreach and does not help advance the state of the science. Please see the following general comments, and line comments.

- *We thank the reviewer for his/her very constructive feedback and detailed assessment of our study. We have addressed all comments below.*

**General comments*:**

Carbon loss from drought - I will start with my strongest objection to this study, which is the extrapolation of forest carbon loss from drought. Accurate estimation of tropical forest carbon loss from drought is a highly sought after goal for tropical ecosystem ecology, but the methods this study uses are not robust or defensible in the present day. The standing biomass and forest sensitivity to drought differs dramatically across Amazonia. This point is even acknowledged (Line 435) in the manuscript. This study does not present any new field data to evaluate this very simplistic empirical relationship (from Lewis 2011), and therefore this study does not have the substance to make these claims. Even Lewis (2011) states this is a first approximation approach and does not include any goodness of fit statistics, the number of plots used to derive this estimate, or even specific information about which RAINFOR plots were included. Lewis extrapolated the relationship beyond the MCWD observed within the RAINFOR plot network from the 2005 drought through the 2010 drought to produce a quick estimate of carbon loss. In this study, the simplistic linear relationship is extrapolated even further beyond the original Lewis 2011 extrapolation. Even if this original relationship was remotely accurate for the 2005 drought, there is no evidence that it was accurate for subsequent droughts in 2010 (or 2015/16). It is difficult to make these forest carbon loss estimates regarding the 2015/16 drought without new field observations and validation, therefore I do not agree that the AGB loss estimates presented here are justifiable and object to their inclusion.

- *We appreciate the reviewers' very comprehensive comments. We agree that we overlooked the study of Feldpausch et a. 2016 which shows that the 2005 AGB-MCWD relationship cannot be applied for 2010 and no evidence exists which would justify the application of the relationship ship for 2015/2016. Hence, we will remove the according estimation of the drought impact on AGB for 2010 and 2015/16, but keep the estimate for 2005. For 2010 and 2015/2016 we will focus on the comparison of the drought indices instead of the AGB estimates.*
- *Generally, the point of our study is not to give better estimates of AGB loss during drought, but rather show how the choice and version of a climate (forcing) dataset also can have large influences on the drought impact and representation. In addition, we wanted to highlight that despite having better satellites and more sophisticated techniques when estimating gridded*

*rainfall maps uncertainty rather increases than decreases. We will add a better explanation about this into the discussion.*

Next, it is worth noting that a large-scale squall line also crossed the Amazon basin during the period of measurements presented in the original Phillips 2009 Science paper. This was estimated to have killed hundreds of millions of trees (Negrón-Juárez et al., 2010 Geophysical Research Letters), so even the empirical AGB~MCWD loss relationship presented in Lewis 2011 has a heavy bias from wind mortality. I strongly urge the authors to drop this aspect of the manuscript. Estimating Amazonian forest carbon loss from drought has long been a difficult endeavour, and many groups have been physically collecting field observations to quantify this. I worry this aspect of the study adds more noise than value to the current state of the science.

- *We thank the reviewer for this detailed comment. As stated in the comment above we agree with the reviewer and will remove the AGB-MCWD relationship for 2010.*

Defining drought - I think the evaluation of different precipitation datasets concerning the drought is mostly fine and could be useful. However the way drought is defined here is a bit simplistic, especially regarding the MCWD anomaly. The mean annual precip spans from 3500 mm + in the northwest Amazon to less than 1700 mm in the southeastern peripheries. I think it is difficult to justify a definition of drought based on absolute thresholds for the MCWD anomaly. The northwest Amazon rarely experiences a dry season, whereas the southeast Amazon does not receive rainfall for more than half the year. Forests are adapted to some level of water stress, which is why simple absolute thresholds are unlikely to characterize vegetation water stress. Assessing drought anomalies based on the number of standard deviations (calculated per pixellocation) is one commonly used way to assess drought with respect to the baseline climate and interannual variability of precipitation. Absolute thresholds (e.g. MCWD >25) vs. relative anomalies (e.g. MCWD > 2 standard deviations). The older papers using MCWD (e.g. Aragão et al., 2007) used a fixed value because there was not enough information at the time of actual ET. Now it is well understood that actual ET can vary substantially across the Amazon and has seasonality in most regions. It no longer makes sense to use a fixed value of ET for both the everwet northwest Amazon and the seasonally dry southwest Amazon. I suggest the authors could use newer spatially resolved ET estimates such as from GLEAM, MODIS MOD16, Fluxcom, etc.

- *While we generally agree with the reviewer that precipitation is very heterogeneously distributed across the Amazon rainforest, the constant ET of 100 is still being used in very recent publications (e.g. Flack-Prain et al. 2019, Biogeosciences; Koch et al. 2021 Earth's Future). As this study was not intended to develop a better drought index, but rather on the comparisons, we would keep the simple ET = 100mm.*
- *We already also did some analyses in the SI using ERA5 ET instead of constant ET=100 which did not affect affected areas too much. We will highlight these differences more in the main study and provide further figures in the SI.*
- *While initially, absolute thresholds are useful for deriving the absolute impact of AGB change, we agree switching to relative thresholds is more meaningful.*

The comparison of precipitation products and drought metrics could be a useful contribution, however this is currently muddled by putting all the estimates together in an ensemble. I suggest the authors focus on presenting a more organized comparison of (1) precipitation products, and (2) drought metrics. What is the justification for using an ensemble of precipitation datasets? Why is this better than using the best evaluated precipitation dataset? Consider the timing of the development of these products. Some of them have been operational for over 20 years. Statistical methods, data assimilation and climate reanalysis models have improved dramatically since then. I think it is difficult to argue that an ensemble method is better, especially when including where a coarse resolution earlier generation product (e.g. GPCC) has as much vote as the latest generation of products (e.g. ERA5, GPM IMERG6).

- *We do not fully understand the reviewers critique regarding our approach. We use the term "ensemble" to reflect a collection of datasets that have overlapping spatial and temporal resolutions. We still consider each dataset individually. As mentioned in the text, the scope of our study was to conduct "a systematic analysis of how the most frequently used precipitation datasets differ regarding the spatial extent, location and severity of recent extreme drought events". Obviously, we were not clear enough about this scope and will make this clearer throughout the text.*

**Other comments**

There are a number of typos in both the main text and figures. Some of these are highlighted in the line comments.
There are far too many acronyms in this manuscript. For example, is CHR really a useful shortening of the CHIRPS? Each new acronym makes the manuscript more difficult to read. I suggest limiting the usage of acronyms to the absolute minimum. Wherever possible, use established acronyms such as TRMM. Making up new acronyms of acronyms (TR6, TR7) is confusing and will not help readers comprehend the manuscript. A manuscript of this length does not need additional acronyms to make it shorter.

- *We fully agree and will use the official acronyms instead of making up new ones.*

**Section comments:**

L30: This should be MCWD > 25 mm, no? Also the climatological mean MCWD across Amazonia is quite large. I don't think it makes sense to use a single value to define drought (~25 mm). MCWD >= 25 mm in the southeast Amazon does not indicate drought.
- *Similarly to Lewis et al. 2011 we wanted to use the negative definition of DeltaMCWD, so in this case DeltaMCWD < -25mm would be correct.*
- *However, as already stated above we agree to switching to relative thresholds and will rephrase this part accordingly.*

L 170: The wet season starts at different times of the year across the Amazon. How is the choice of starting the hydrological year determined?

- *Similar to Phillips et al. 2009 and Lewis et al. 2011 we selected the 1st October as the onset of the hydrological year for each location in the Amazon. We will add this more clearly to the methods.*

L 173: I am not sure Delta MCWD is a good abbreviation for the anomaly of MCWD. This can easily be taken as just the change in MCWD between two time periods, but that's not exactly what the anomaly is during a drought. Perhaps it's better to spell it out as the " MCWD anomaly".

- *We thank the referee for this suggestion and will use the term MCWD anomaly throughout the text and figures.*

L 176: Removing the drought years causes bias. There are three droughts in the span of 15 years, so these are not rare events. Just because Lewis 2011 used a method, does not mean it is defensible in the present day.
- *see subsequent comment*

L 185: Climatologies are typically calculated from 30 year periods. Most of the data products have at least 20 years of duration, if not closer to 40. The selection of years to remove is subjective and removing the years with anomalously low rainfall will bias the standard deviation to be artificially small.

- *Regarding L176 and L185, we agree with the reviewer that removing the 3 extreme events may cause a bias. We will include the baseline years 2005, 2010 and 2016 in our MCWD calculation to avoid this bias.*

L208: Be consistent in treating MCWD as either a positive or negative quantity.
- *We will correct any inconsistent use of MCWD throughout the manuscript.*

L215+: I reject the underlying basis for the empirical carbon loss estimate from Lewis (2011).
- *We accept this rejection and do not give estimates of carbon losses for years other than 2005.*

L229: MCWD is misspelled L295: It is difficult for rainfall products to correctly estimate rainfall near the foothills of the Andes. Also some areas have very little ground information for each product's bias correction algorithm. It might be worth getting into this to describe more deeply why the products disagree, and where.
- *We thank the reviewer for mentioning this very important point. We will get into more detail at this point about why datasets disagree near the Andes.*

L333: I would note that many studies no longer use the fixed estimate of 100 mm. I believe some have used Stephenson (1998 Journal of Biogeography) as a reference for the development of the MCWD metric.
- *We will mention Stephenson (1998 Journal of Biogeography) as a reference for MCWD and we will also mention that there are better alternatives for ET instead of assuming constant ET =*

*100mm/month. However, there are still quite some studies that are using the constant ET=*
*100mm/month approximation which we will also highlight at this point.*

L357: Using a better estimate of "actual ET" might reflect the impact of VPD. I would say this is a limitation of using a fixed 100mm value for ET in the MCWD calculation.

- *This is a good point and we will include this in our discussion.*

L426: Indeed, this is another reason to drop the extrapolated carbon loss estimates.

- *See above sections.*

L453: I don't think the case for assessing drought with an ensemble is made clear.

- *We will rephrase this to make it more clear (see also comment below)*

Why is it not better to just use the product that has the lowest RMSE in the region of interest?
- *We again want to highlight that the purpose of this study is not to find the best dataset for locations at which we have exact measurements and can evaluate RMSE, but to give a broad picture of how different precipitation datasets represent drought stress across the complete basin. We will rephrase some parts of the manuscript to make this more clear.*

L458: The code in the repo looks to be incomplete. Ideally the complete code for analysis and figures should be hosted prior to the review process. An incomplete repository hinders the review process.
- *We will put all the files and scripts on the repository so that they can be easily reproduced.*

Figure 1: Is this MCWD, or anomalies of MCWD?
Figure 2: Why is WAT not included in panel C?
Figure 3: This is a useful figure. It might be useful to add another two columns indicating where the satellite based products agree, and where the climate reanalysis modeled products agree.
Figure 4: Is "PA" (y-axis label) supposed to be "RAI"?
Figure 5: Is "PA" (y-axis label) supposed to be "RAI"? The delta MCWD supposed to be the Anomaly of MCWD? Might be better to spell this out.
Figure 6: I suggest removing this aspect of the study, and this figure.
Table 1: I suggest dropping the abbreviations of abbreviations, and adding a column about how the product is derived (e.g. Remote sensing, interpolation of ground data, atmospheric process model, etc).
Table 2: RAI?
- *We thank the reviewer for the in-depth checking of our figures and will fix all the listed issues.*

---

## Author Response (AR1)

**Response Reviewer 1**

**General comments*:**

Papastefanou et al. assessed the extent and severity of the 2005,2010, and 2015/2016 droughts over the Amazon basin using 10 precipitation data sources and 3 drought indexes (MCWD, scPDSI, and RAI) with different assumptions. The main results show an increasing disagreement across datasets for more severe drought signals (in terms of both frequency and location). PDSI which consider variable ET shows a much stronger drought impact in 2016 compared with MCWD while RAI based on dry season rainfall shows a weaker drought impact in 2016. In addition, the research explored the consequences of estimating biomass loss from uncertainty across different precipitation using an empirical drought-mortality relationship. The resultant uncertainty in total carbon loss can reach 1.4 PgC (1.3-2.7) for the 2015/2016 drought. The authors conclude with a recommendation of using an ensemble of precipitation data sets when assessing the impact of drought. Overall, I think the analysis is a useful contribution to the study of drought impacts over the Amazon or more generally the tropical forests. The research provides a comprehensive overview of the differences across rainfall datasets, an issue that any analysis or modeling studies over tropical drought will struggle with. I feel the key figures showing dataset agreement are helpful. However, I think the manuscript can benefit from more in-depth discussion and a stronger conclusion. Please see the below specific comments for details. Hopefully, they will help to improve the manuscript and make it more useful to the scientific community.

- *We thank the reviewer for his constructive feedback. We addressed all comments in detail in the sections below.*

**Specific Comments*:**

1. The manuscript focuses on the disagreement among drought indices across different precipitation data sets, which are ultimately driven by the differences in precipitation. It would be helpful to show the difference (e.g. systematic biases and spatialtemporal correlation) across the raw precipitation data sets using paired scatter plots for each precipitation data combination (could be put in the supplementary). This can help to understand why there are disagreements in MCWD (is it just because of a systematic bias so certain data set generates lower MCWD or due to disagreement in the spatial distribution of rainfall, etc.) Such analyses can help to illustrate.

- *We agree with the reviewer that analyzing the precipitation datasets in more detail will improve the understanding of the differences of the MCWD.*
- *We added an additional plot (Fig. S1) that shows the empirical cumulative density functions (CDFs) of monthly precipitation. We find no obvious biases between the datasets, with only ERA5 showing consistently higher rainfall rates.*
- *We further created empirical CDFs for each drought index and across all grid cells (Fig. S2). We could, however, also not identify any obvious biases between the precipitation datasets. By comparing the CDFs we were able to express our absolute MCWD anomaly classifications with relative MCWD anomaly classifications (Methods S1) which we further used throughout the manuscript. Using the relative anomalies also enabled us to better cross-compare the three drought indices.*
- *We now refer to the additional analyses in the main text in lines 171- 174 and 382-384.*

A related point is how to compare different drought indices. Current categorization into moderate, severe, and extreme seems too subjective. Why not show the scatter plot between different drought indices across the drought (from selected precipitation dataset or averaged across all precipitation datasets), which can show the scaling between MCWD, scPDSI, and RAI and demonstrates their differences. Or maybe use percentile (e.g. lowest 5% to indicate extreme) to compare across indices?

- *We agree with the reviewer that our categorization is subjective. This was also pointed out by reviewer 2. We refactored figure 1-3 and 5,6 and now use relative MCWD anomaly (in units of standard deviations) to better describe the agreement of the datasets. We only used absolute MCWD anomalies for Fig. 6 (now figure 4) and the potential biomass losses for 2005.*
- *We have now adjusted the text on multiple occasions, e.g. in lines 217, and 269 - 274 where we now write: "Because no relationship between the anomalies of aMCWD and aAGB could be verified for 2010 (Feldpausch et al., 2016) we did not estimate the impacts on AGB for the other drought years 2010 and 2016."*

2. I like the idea of translating uncertainty in MCWD into the uncertainty in AGB changes (ln 215). However, it should be acknowledged that the empirical relationship itself subjects to large uncertainty. For example, Feldpausch et al. (2016) find that the mortality-MCWD relationship identified in 2005 disappeared during the 2010 drought. Feldpausch T R, Phillips O L, Brienen R J W, Gloor E, Lloyd J, Lopez-Gonzalez G, Monteagudo-Mendoza A, Malhi Y, Alarcón A, Álvarez Dávila E, Alvarez-Loayza P, Andrade

A, Aragao L E O C, Arroyo L, Aymard C. G A, Baker T R, Baraloto C, Barroso J, Bonal D, Castro W, Chama V, Chave J, Domingues T F, Fauset S, Groot N, Honorio Coronado E, Laurance S, Laurance W F, Lewis S L, Licona J C, Marimon B S, Marimon-Junior B H, Mendoza Bautista C, Neill D A, Oliveira E A, Oliveira dos Santos C, Pallqui Camacho N C, Pardo-Molina G, Prieto A, Quesada C A, Ramírez F, Ramírez-Angulo H, Réjou-Méchain M, Rudas A, Saiz G, Salomão R P, Silva-Espejo J E, Silveira M, ter Steege H, Stropp J, Terborgh J, Thomas-Caesar R, van der Heijden G M F, Vásquez Martinez R, Vilanova E and Vos V A 2016 Amazon forest response to repeated droughts Global Biogeochem. Cycles 30 964–82 Online: https://agupubs.onlinelibrary.wiley.com/doi/full/10.1002/2015GB005133 In addition, I am not sure whether directly plugging in MCWD based on different rainfall data set makes sense. eqn 2 was derived using a specific rainfall data set. I think it would make more sense to remove the systematic biases between the specific data set and all the data set used in this study before converting MCWD to AGB. One way to find the mapping between MCWD data sets is simple regressions between the data sets as suggested in my comment above. Will such cross-data set calibration reduce AGB uncertainty?

- *We thank the reviewer and author for highlighting the Feldpausch et al. 2016 study which we missed when writing our manuscript and we agree that the linear relation between AGB and MCWD does not hold for 2010 and 2016. This was also pointed out by reviewer 2. We removed the AGB loss estimates for 2010 and 2016.*
- *We appreciate the suggestion of the referee to deeper investigate the MCWD-AGB relation using multiple precipitation datasets and we would be happy to work on this topic together in a follow-up study.*

3. Current conclusion recommends using an ensemble of different rainfall data sets when analyzing drought impacts. However, is there strong evidence that the ensemble would perform better than individual data sets? I wonder whether there are ways to evaluate the performance of each rainfall data set in terms of estimating drought impact. For example, is it possible to compare the spatial and temporal patterns of AGB loss based on different rainfall data sets with the observed spatial-temporal patterns from microwave remote sensing data (Liu et al. 2015; Saatchi et al. 2013; Wigneron et al. 2020) or lidar data (Yang et al. 2018)? Some more detailed details on the potential biases of MCWD that do not include ET variability?

- *We thank the referee for these important remarks. While we do not want to state that an ensemble (collection of datasets) generally performs better than one single dataset, our point is that drought stress can differ substantially between datasets. So for studies assessing impacts of droughts on the Amazon rainforest it may be worth considering multiple datasets to test for climate uncertainty purely arising by the choice of precipitation dataset. We reformulated our manuscript accordingly, it now reads in lines 36-37, 455-456 and 470-472: "Communicating the uncertainty in the estimation of drought events and their impacts on the Amazon rainforest is highly relevant and thus, multiple datasets should be applied by any large-scale study on drought impacts on vegetation."*
- *We appreciate the reviewers' idea regarding the comparisons to remotely sensed AGB data. While this would probably go beyond the scope of this study we think that it would be interesting to investigate in a follow-up study.*

In 369, I thought microwave data is mostly free from cloud cover effect, which mainly influence optical remote sensing products? I think some of the challenges are the limited penetration depth in the dense tropical forests (Chaparro et al. 2019) and the influences of vegetation water status (Xu et al. 2021)

- The reviewer is correct. We fixed the sentence and thank the reviewer for pointing out some challenges of microwave data which we included in the text: "However, conducting analyses over the Amazon rainforest based on VOD is difficult, because of the limited penetration depth of microwaves in dense tropical forests (Chaparro et al. 2019), and the influences of vegetation water status (Xu et al. 2021)." (Lines 372-374)

**Stylistic Comments and Technical Corrections:**

ln 63: 'altering the carbon cycle of the Amazon forest already today' -> 'already altering the carbon cycle of the Amazon forest'
- We fixed this styling issue.

ln 80-100: I wonder whether it is better to just briefly talk about the usage of ten different data sets here and move the details into Methods

- We acknowledge that this part of the introduction might be long, but we think that the details presented are useful in the introduction as they give the reader a short overview of the datasets used in this study. Hence, we would like to keep this part in the introduction.

ln 122: 0.6 Mio -> 0.6 million?
- Fixed!

ln 402: 'In addition, also', the also is extra
- Fixed!

ln 419: 'average annual carbon uptake' global or regional? Please specify I wonder whether Table 2 and Table 3 are more suitable for SI... Especially if additional figures on the difference across rainfall datasets are added in the revision.
- We thank the reviewer for his suggestion and moved Table 2 and Table 3 to the supporting information, they are now Table S2 and S3, their reference was updated throughout the manuscript text.

**Response Reviewer 2**

The authors present a comparison of drought metrics, calculated with different rainfall products. The study region is focused on the Amazon basin, and an extrapolation is made of aboveground forest carbon loss from drought. The authors end with a message that evaluation of drought through an ensemble is better. I think the comparison of rainfall products and evaluation of drought metrics could be useful, especially if it is more developed in the revision. This section could use some more analysis, especially with respect to defining anomalies per pixel location rather than absolute thresholds. However the section concerning the extrapolation of forest carbon loss from drought is a large overreach and does not help advance the state of the science. Please see the following general comments, and line comments.

- *We thank the reviewer for his/her very constructive feedback and detailed assessment of our study. We have addressed all comments below.*

**General comments*:**

Carbon loss from drought - I will start with my strongest objection to this study, which is the extrapolation of forest carbon loss from drought. Accurate estimation of tropical forest carbon loss from drought is a highly sought after goal for tropical ecosystem ecology, but the methods this study uses are not robust or defensible in the present day. The standing biomass and forest sensitivity to drought differs dramatically across Amazonia. This point is even acknowledged (Line 435) in the manuscript. This study does not present any new field data to evaluate this very simplistic empirical relationship (from Lewis 2011), and therefore this study does not have the substance to make these claims. Even Lewis (2011) states this is a first approximation approach and does not include any goodness of fit statistics, the number of plots used to derive this estimate, or even specific information about which RAINFOR plots were included. Lewis extrapolated the relationship beyond the MCWD observed within the RAINFOR plot network from the 2005 drought through the 2010 drought to produce a quick estimate of carbon loss. In this study, the simplistic linear relationship is extrapolated even further beyond the original Lewis 2011 extrapolation. Even if this original relationship was remotely accurate for the 2005 drought, there is no evidence that it was accurate for subsequent droughts in 2010 (or 2015/16). It is difficult to make these forest carbon loss estimates regarding the 2015/16 drought without new field observations and validation, therefore I do not agree that the AGB loss estimates presented here are justifiable and object to their inclusion.

- *We appreciate the reviewers' very comprehensive comments. We agree that we overlooked the study of Feldpausch et a. 2016 which shows that the 2005 AGB-MCWD relationship cannot be applied for 2010 and no evidence exists which would justify the application of the relationship for 2015/2016. Hence, we removed the impact of the droughts on AGB for 2010 and 2015/16, but kept the estimate for 2005. For 2010 and 2015/2016 we focused on the comparison of the drought indices instead of the AGB estimates. We adapted the method section and wrote "To calculate the AGB anomaly in Eq. 2, we calculated the MCWD anomaly of each gridcell in 2005 for each of the precipitation datasets in our study." (lines 216-217) and also the result and discussion sections in multiple occasions, e.g. in lines (272-274) where we wrote "Because no relationship between the anomalies of MCWD and AGB could be verified for 2010 (Feldpausch et al., 2016) we did not estimate the impacts on AGB for the other drought years 2010 and 2016."*
- *Generally, the point of our study is not to give better estimates of AGB loss during drought, but rather to show how the choice and version of a climate (forcing) dataset also can have large influences on the drought impact and representation. In addition, we wanted to highlight that despite having better satellites and more sophisticated techniques the uncertainty can even increase for recent drought events (such as 2016). We added a better explanation about this into the discussion and conclusion e.g. in lines 453-455 and 468-470: "We therefore recommend applying several climate (precipitation) datasets as well as drought metrics to account for model uncertainty when assessing the spatial extent, duration, and location of droughts. We regard it as an important step when assessing drought impacts on tropical rainforests also under current climate conditions."*

Next, it is worth noting that a large-scale squall line also crossed the Amazon basin during the period of measurements presented in the original Phillips 2009 Science paper. This was estimated to have killed hundreds of millions of trees (Negrón-Juárez et al., 2010 Geophysical Research Letters), so even the empirical AGB~MCWD loss relationship presented in Lewis 2011 has a heavy bias from wind mortality. I strongly urge the authors to drop this aspect of the manuscript. Estimating Amazonian forest carbon loss from drought has long been a difficult endeavour, and many groups have been physically collecting field observations to quantify this. I worry this aspect of the study adds more noise than value to the current state of the science.

- *We thank the reviewer for this detailed comment. As stated in the comment above we agree with the reviewer and removed the AGB-MCWD relationship for 2010 and 2016 from figure 6 (now figure 4). Figure 4 is now:*

[Figure]

Defining drought - I think the evaluation of different precipitation datasets concerning the drought is mostly fine and could be useful. However the way drought is defined here is a bit simplistic, especially regarding the MCWD anomaly. The mean annual precip spans from 3500 mm + in the northwest Amazon to less than 1700 mm in the southeastern peripheries. I think it is difficult to justify a definition of drought based on absolute thresholds for the MCWD anomaly. The northwest Amazon rarely experiences a dry season, whereas the southeast Amazon does not receive rainfall for more than half the year. Forests are adapted to some level of water stress, which is why simple absolute thresholds are unlikely to characterize vegetation water stress. Assessing drought anomalies based on the number of standard deviations (calculated per pixellocation) is one commonly used way to assess drought with respect to the baseline climate and interannual variability of precipitation. Absolute thresholds (e.g. MCWD >25) vs. relative anomalies (e.g. MCWD > 2 standard deviations). The older papers using MCWD (e.g. Aragão et al., 2007) used a fixed value because there was not enough information at the time of actual ET. Now it is well understood that actual ET can vary substantially across the Amazon and has seasonality in most regions. It no longer makes sense to use a fixed value of ET for both the everwet northwest Amazon and the seasonally dry southwest Amazon. I suggest the authors could use newer spatially resolved ET estimates such as from GLEAM, MODIS MOD16, Fluxcom, etc.

- *Thank you very much for raising this point. While we generally agree with the reviewer that precipitation is very heterogeneously distributed across the Amazon rainforest, the constant ET of 100mm is still being used frequently in recent publications (e.g. Flack-Prain et al. 2019, Biogeosciences; Koch et al. 2021 Earth's Future). While we developed the methods for this study we also looked into this topic acknowledging the spatially large difference in annual precipitation.*
- *We already also did some analyses (Fig. S3) where we tested the sensitivity of the 100 mm ET threshold and used the ERA5 ET product instead. We found that using variable*

*ET can significantly reduce the MCWD anomalies. However as (to our knowledge) the majority of the studies conducted in the Amazon rainforest still use constant ET of 100mm we would like to keep the 100mm ET threshold.*

- *To account for this important effect of variable ET on MCWD we added the following sentence to the discussion in line 335-340: "We investigated the effect of choosing variable evapotranspiration and a longer baseline in our MCWD calculation (Fig. S3). Using variable evapotranspiration consistently reduced the moderate drought-affected area by 10-20% per drought event (Fig. 3a, b, c). It also affected the intensity of the drought stress, e.g. areas previously classified as extreme drought affected were now classified as areas with severe drought stress. This reduction is expected as ERA5 takes the above-mentioned lower ET values in the highland tropics into account which overall leads to higher MCWD values in this region. Because of the strength and consistency of this effect we recommend testing the MCWD calculation regarding its sensitivity to variable ET in the tropical rainforest in future studies."*
- *While initially, absolute thresholds are useful for deriving the absolute impact of AGB changes, we agreed switching to relative thresholds throughout the study is more meaningful.  This further enabled a better cross-comparison of MCWD to the two other indices which we now compared directly. We hope with the current changes on the use of the MCWD we have fully addressed this point.*

The comparison of precipitation products and drought metrics could be a useful contribution, however this is currently muddled by putting all the estimates together in an ensemble. I suggest the authors focus on presenting a more organized comparison of (1) precipitation products, and (2) drought metrics. What is the justification for using an ensemble of precipitation datasets? Why is this better than using the best evaluated precipitation dataset? Consider the timing of the development of these products. Some of them have been operational for over 20 years. Statistical methods, data assimilation and climate reanalysis models have improved dramatically since then. I think it is difficult to argue that an ensemble method is better, especially when including where a coarse resolution earlier generation product (e.g. GPCC) has as much vote as the latest generation of products (e.g. ERA5, GPM IMERG6).

- *We are not quite sure if we fully understand the reviewers critique regarding our approach. We use the term "ensemble" to reflect a collection of datasets that have overlapping spatial and temporal resolutions. The collection of climate forcing data sets that we use include state-of-the-art reanalysis data sets (NCEP, ERA5 and 20CR), remotely sensed data sets (TRMM v6 and v7), widely used climatology data sets which are interpolated from station data (CRU) and merged data sets (GSWP3 and WATCH-*

*WDFEI). Our aim is to show the range of climate forcings and the resulting simulated ET and drought response. We therefore analysed the time period which is covered by all data sets. However, we still consider each dataset individually as shown, e.g., in Fig. 2. As mentioned in the text (lines 75 - 76), the scope of our study was to conduct "a systematic analysis of how the most frequently used precipitation datasets differ regarding the spatial extent, location and severity of recent extreme drought events". Obviously, we were not clear enough about this scope and tried to make this clearer throughout the text, for example in lines 36 to 37, where we now write: "We conclude that for deriving impacts of droughts on the Amazon Basin based on precipitation, multiple datasets should be considered."*

**Other comments**

There are a number of typos in both the main text and figures. Some of these are highlighted in the line comments.

There are far too many acronyms in this manuscript. For example, is CHR really a useful shortening of the CHIRPS? Each new acronym makes the manuscript more difficult to read. I suggest limiting the usage of acronyms to the absolute minimum. Wherever possible, use established acronyms such as TRMM. Making up new acronyms of acronyms (TR6, TR7) is confusing and will not help readers comprehend the manuscript. A manuscript of this length does not need additional acronyms to make it shorter.

- *We fully agree and used the official acronyms throughout the manuscript instead of making up new ones.*

**Section comments:**

L30: This should be MCWD > 25 mm, no? Also the climatological mean MCWD across Amazonia is quite large. I don't think it makes sense to use a single value to define drought (~25 mm). MCWD >= 25 mm in the southeast Amazon does not indicate drought.

- *Similar to Lewis et al. 2011 we wanted to use the negative definition of DeltaMCWD, so in this case DeltaMCWD < -25mm would be correct.*
- *However, as already stated above we switched to relative thresholds and rephrased this part accordingly. It now reads (lines 24 to 26): "Evaluating an ensemble of nine state-of-the-art precipitation datasets for the Amazon region, we find that the spatial extent of the drought in 2005 ranges from 2.2 to 3.0 (mean = 2.7) million km² (37 – 51% of the Amazon basin, mean = 45%) where MCWD indicates at least moderate drought conditions (relative MCWD anomaly < -0.5).".*

L 170: The wet season starts at different times of the year across the Amazon. How is the choice of starting the hydrological year determined?

- *Similar to Phillips et al. 2009 and Lewis et al. 2011 we selected the 1st October as the onset of the hydrological year for each location in the Amazon.*

L 173: I am not sure Delta MCWD is a good abbreviation for the anomaly of MCWD. This can easily be taken as just the change in MCWD between two time periods, but that's not exactly what the anomaly is during a drought. Perhaps it's better to spell it out as the " MCWD anomaly".

- *We thank the referee for this suggestion and now use the term MCWD anomaly throughout the text and figure descriptions.*

L 176: Removing the drought years causes bias. There are three droughts in the span of 15 years, so these are not rare events. Just because Lewis 2011 used a method, does not mean it is defensible in the present day.
- *Please see subsequent comment because we think they are related.*

L 185: Climatologies are typically calculated from 30 year periods. Most of the data products have at least 20 years of duration, if not closer to 40. The selection of years to remove is subjective and removing the years with anomalously low rainfall will bias the standard deviation to be artificially small.

- *Regarding L176 and L185, we agree with the reviewer that removing the 3 extreme events may cause a bias. We included the baseline years 2005, 2010, and 2016 in our MCWD calculation (and also in the calculation of the other metrics) to avoid this potential bias.*

L208: Be consistent in treating MCWD as either a positive or negative quantity.
- *We corrected any inconsistent use of MCWD throughout the manuscript.*

L215+: I reject the underlying basis for the empirical carbon loss estimate from Lewis (2011).
- *We accepted this rejection and deleted all estimates of carbon losses for years other than 2005.*

L229: MCWD is misspelled L295: It is difficult for rainfall products to correctly estimate rainfall near the foothills of the Andes. Also, some areas have very little ground information for each product's bias correction algorithm. It might be worth getting into this to describe more deeply why the products disagree, and where.

- We corrected the spelling error, thanks for spotting it.
- Also, thanks for raising the issue on estimating rainfall at the Andean foothills correctly. Interestingly, when switching to relative anomalies, as suggested by the reviewer above, this disagreement at the Andes disappeared.  This also supports the reviewer's suggestion of using relative over absolute thresholds.
- While switching to relative anomalies the most of the more obvious differences (e.g. in the highlands) disappeared. Furthermore,  we could not find much in the literature that could explain the differences we observed in our studies. An in-depth analysis why the forcings disagree would probably go beyond the scope of this study.

L333: I would note that many studies no longer use the fixed estimate of 100 mm. I believe some have used Stephenson (1998 Journal of Biogeography) as a reference for the development of the MCWD metric.

- *We mentioned Stephenson (1998 Journal of Biogeography) as a reference for MCWD in line 319.*
- *We now mention that there are better alternatives for ET instead of assuming constant ET = 100mm/month in lines 330 to 332, where we state: "In the last decade, better products of spatially and temporally resolved evapotranspiration data (e.g. ERA5) have been developed and an increasing number of studies are now estimating MCWD based on such data (e.g. Staal et al., 2020)".*
- *However, we also pointed out that many studies (e.g. Flack-Prain et al., 2019; Koch et al., 2021)  in the Amazon still use the constant ET= 100mm/month approximation. See also our above response where we addressed the major points.*

L357: Using a better estimate of "actual ET" might reflect the impact of VPD. I would say this is a limitation of using a fixed 100mm value for ET in the MCWD calculation.

- *We thank the reviewer for this good point and added this point to our discussion in line 366 where we state: "One possibility to account for the influences of VPD is choosing temporal and spatially resolved evapotranspiration instead of constant evapotranspiration in the calculation of MCWD."*

L426: Indeed, this is another reason to drop the extrapolated carbon loss estimates.

- *See our response to the first major point (above).*

L453: I don't think the case for assessing drought with an ensemble is made clear.

- *We rephrased this part to make it more clear (see also comment below). It now reads (lines 454 to 455): "Therefore, we recommend using multiple climate forcing datasets to test for climate data uncertainty also under present climate conditions.".*

Why is it not better to just use the product that has the lowest RMSE in the region of interest?

- *We again want to highlight that the purpose of this study is not to find the best dataset for locations at which we have exact measurements and can evaluate RMSE, but to give a broad picture of how different precipitation datasets represent drought stress across the complete basin. We rephrased (e.g. in the introduction and conclusions) some parts of the manuscript to make this more clear. For example, we now state in the introduction (lines 107 to 111): "The goals of our study are (1) to analyze and quantify the uncertainty in strength, extent, and location of three recent Amazon droughts in the years 2005, 2010, and 2015/2016 in precipitation from nine state-of-the-art precipitation or climate datasets based on MCWD; (2) to examine differences among these drought events by taking two additional drought indicators RAI and scPDSI into account; and (3) to give an estimate of the impacts of the 2005 drought on the carbon cycle by estimating potential biomass losses."*

L458: The code in the repo looks to be incomplete. Ideally the complete code for analysis and figures should be hosted prior to the review process. An incomplete repository hinders the review process.

- *We will put all the files and scripts on the repository so that they can be easily reproduced.*

Figure 1: Is this MCWD, or anomalies of MCWD?

- We replaced Figure 1 with a new figure representing the impacts of the 2016 drought across all datasets. Figure 1 now is:

[Figure]

Figure 1: Relative MCWD anomalies (from October to September) as an indicator for drought stress in the Amazon basin during the record-breaking drought event in 2016. Displayed are only the datasets that include the year 2016 in their temporal range. The baseline period of the MCWD calculation is 2001 to 2016.

Figure 2: Why is WAT not included in panel C?
- Because WATCH_WFDEI (or at least the version we have of it) does not go beyond 2010.

Figure 3: This is a useful figure. It might be useful to add another two columns indicating where the satellite-based products agree, and where the climate reanalysis modeled products agree.
- We thank the reviewer for finding this figure useful. We agree that it would be interesting to compare against satellite-based products. However, this would go beyond the scope of this study as our purpose was not to find the "best performing" dataset, but rather to show the uncertainties that purely arise by the selection of a dataset (and drought indices).

Figure 4: Is "PA" (y-axis label) supposed to be "RAI"?
- This is correct and we fixed it.

Figure 5: Is "PA" (y-axis label) supposed to be "RAI"? The delta MCWD supposed to be the Anomaly of MCWD? Might be better to spell this out.
- This is correct and we fixed it.

Figure 6: I suggest removing this aspect of the study, and this figure.
- (Now Figure 4). We removed the 2010 and 2016 aspects from this figure. However, we still would like to keep this figure and also to show the differences of the potential impacts on the carbon balance for 2005, where the Lewis et al. 2011 (Phillips et al. 2008) relationship is valid.

Table 1: I suggest dropping the abbreviations of abbreviations, and adding a column about how the product is derived (e.g. Remote sensing, interpolation of ground data, atmospheric process model, etc).
- Thank you for this suggestion. We dropped the abbreviations of abbreviations from the table and added the column about how the product is derived.

Table 2: RAI?
- Correct and fixed. Please note that we moved the big tables 2 and 3 to the supporting information to follow a suggestion from reviewer 1.

---

## Author Response (AR2)

Reviewer #1

General comments:

First, kudos to the authors for publishing their code. This is good practice and helps ensure reproducibility.

*We thank the reviewer for this comment and think that code should always be made publicly available.*

I am not asking the authors to do this, but the other equally important dimension of calculating MCWD is the estimate of ET. Obviously this can produce large differences, so I don't quite understand why this was not considered. These different combinations of precipitation and ET products could produce some very contrary estimates of MCWD. I think this would be a very citeable finding that this analysis is well suited to do, and it would be a very useful contribution to the literature. However I understand if this is infeasible, and the discussion of ET differences in the discussion is useful.

We thank the reviewer for this very good suggestion and now included both ET datasets (DOLCE and GLEAM) that the reviewer suggested. We extended our analysis of the sources of variability from precipitation datasets and drought indices to now also include the variability caused by the choice of evapotranspiration dataset. We further also compare and quantify the differences when using variable ET from DOLCE and GLEAM ET against the fixed ET= 100mm per month assumption (new Fig. 4). Thereby, we find an overestimation of drought stress for all the three drought years 2005, 2010, 2016 when using a fixed ET of 100 mm per month. This overestimation gets more pronounced the further South the drought is located. We rewrote parts of the methods, results and discussion sections in the light of these new analyses.

The calculation of relative MCWD anomalies is a bit confusing. I did not understand when the 10 year interval was used to calculate the baseline, and when 16 years was used (L165). Ten or even 16 years is short for a reference period, which is typically closer to 30 years.

We used 10 years for the 2005 and 2010 drought events and 16 years for the 2016 event. We did so, because some datasets, like TRMM v6 and GSWP3 end in 2010 and others, such as TRMM v7, only start in 1999.

The discussion around differences in precipitation products is useful (as is Figure 3), but I wonder if this analysis (or discussion) could probe deeper into why these products disagree in some areas. Is it because of differences in ground station locations used by the products? Is it because some only use infrared data, and others incorporate microwave soundings?

We added some more sentences about the differences between the precipitation datasets to the discussion (e.g. lines…) and now also mention bias-correction as another source that introduces differences across the precipitation datasets. We agree that it would be very interesting to go even deeper and explore if we can find any patterns of methodological origin that can better explain the differences between the datasets. However, because of the complexity (climate models vs. satellite observations, reanalysis, bias-correction, etc…) with which such datasets have been created we argue that this would rather require a dedicated study itself. The precipitation datasets used for this study are very independent (see Table 1) and therefore there is not a surprise that they differ substantially

even at a global scale (see e.g. Figure 2.15 Gulev et al., 2021), and even less surprising on this regional scale (Doblas-Reyes et al., 2021). This is why it is so important to take into account the observational uncertainty in regional climate studies. For example, four of the products are based on different reanalysis - these are four different Global Climate Models that assimilate observed data during execution. The simulatedprecipitation fields of ERA5 are not bias-corrected while NCEP-CRU and WATCH_WFDEI are bias-corrected with the gridded product CRU while GSWP3 is corrected with the gridded product GPCC. Even CRU and GPCC can give very different results at a regional scale (see figures 10.12 and 10.13 in Doblas-Reyes et al., 2021). Similarly, the products CHIRPS and TRMM are not based on comprehensive global climate models, but on satellite data that use different instruments and retrieval models (TRMM and CHIRPS), CHIRPS is further merged with observed in-situ data.

Concerns:

My biggest concern is regarding the simplistic estimation of AGB loss. I am disappointed to see the authors did not accept my earlier recommendation to drop this. The Lewis et al., (2011) paper managed to get an estimate based on a simple one term regression with a lot of actual forest inventory data, but this does not mean this is a robust way to estimate carbon loss. It does not make sense that the same MCWD value would cause equivalent loss of AGB across Amazonia when the baseline carbon stocks are different, and the forests are adapted to different seasonal variations of MCWD (i.e. aseasonally wet northwest vs seasonally dry southeast). It is not surprising that somewhat different numbers will be generated from this (flawed) approach (Figure 4), and I worry that we will see more of this approach if I were to accept this. Again, I ask the authors to remove this part of the manuscript. I think the rest of the manuscript is acceptable, but not this estimate of AGB carbon loss.

We agree with the reviewer that it is probably not feasible to apply the MCWD-AGB from Lewis et al. 2011 to the other MCWD estimates of our study. We dropped Figure 4 and all MCWD-AGB related estimates from our study. We still want to highlight that the goal of this study was not to give better estimates of the drought impact, but rather highlight the differences that arise by purely choosing a different precipitation dataset.

The sections in the Results about the calculation of the RAI and scPDSI should include more specific details about how these indices are actually calculated. It would be more clear to list the equations. Also I don't think equation 1 is quite correct as the WD or CWD is constrained to always be ≤ 0.

The reviewer is correct. We modified equation 1 accordingly:

if (P(t) - ET(T) < 0)

WD(t) = P(t) - ET(T)

else

WD(t) = 0

The ERA5 PET has a known bug: "The Potential Evaporation field (pev, parameter Id 228251) is largely underestimated over deserts and high-forested areas. This is due to a bug in the code that does not allow transpiration to occur in the situation where there is no low vegetation." from https://confluence.ecmwf.int/display/CKB/ERA5%3A+data+documentation.

Even the ERA5-Land PET seems problematic. Perhaps it would be better to derive a monthly climatology from GLEAM (https://www.gleam.eu/), or use one of the recent multi-product merges such as the newer version of DOLCE (https://hess.copernicus.org/articles/22/1317/2018/). I mention using a climatology of ET instead of actual monthly estimates of ET (or PET) because it would account for seasonal variation (L325), but also because the error of any ET product is likely to be very large with potentially spurious seasonal patterns.

We thank the reviewer for this detailed critique regarding PET data sets. We removed ERA-Land PET from our study and included GLEAM and DOLCE in our study.

Figure 4: I strongly suggest removing this figure.

We removed this figure from our study and included a new figure 4 (see response to your comments above).

Figure 6: This is a useful figure but these colors (red and green) are not distinguishable by colorblind people. Yellow on white is also difficult to distinguish.

We thank the reviewer for the close look and chose a different color scheme for figure 6.

**Literature**

Doblas-Reyes, F. J., Sorensson, A. A., Almazroui, M., Dosio, A., Gutowski, W. J., Haarsma, R., Hamdi, R., Hewitson, B., Kwon, W.-T., Lamptey, B. L., Maraun, D., Stephenson, T. S., Takayabu, I., Terray, L., Turner, A., & Zuo, Z. (2021). *Linking global to regional climate change* (V. Masson-Delmotte, P. Zhai, A. Pirani, S. L. Connors, C. Pean, S. Berger, N. Caud, Y. Chen, L. Goldfarb, M. I. Gomis, M. Huang, K. Leitzell, E. Lonnoy, J. B. R. Matthews, T. K. Maycock, T. Waterfield, O. Yelekci, R. Yu, & B. Zhou, Eds.). Cambridge University Press. https://centaur.reading.ac.uk/99896/

Gulev, S., Thorne, P., Ahn, J., Dentener, F., Domingues, C., Gerland, S., Gong, D., Kaufman, D., Nnamchi, H., Quaas, J., Rivera, J., Sathyendranath, S., Smith, S., Trewin, B., Schuckmann, K., & Vose, R. (2021). *IPCC AR6 WGI Chapter 2: Changing state of the climate system*.

Reviewer #2

The revision addresses some of the major comments in the first round (e.g. questions on extrapolating the drought-mortality relationship derived from the 2005 drought to 2010 and 2015/2016) and improves the clarity and accuracy of the manuscript. However, I feel one of my major comment about why the drought intensity differs was not fully answered.

First, I was suggesting pair-wise scatter plots (or heat maps) between MCWD generated from all data sets. Such figures common for all inter-comparison studies and accompanying regression analyses can tell the spatio-temporal correlation (R2) and systematic biases (intercept and slope). In the revision, the authors present a comparison of CDFs, which are very qualitative and do not contain the spatio-temporal structure as in a scatter plot. I am still suggesting the inclusion of such pair-wise comparisons (either using scatter plots or just reporting correlation/regression statistics) instead of comparing CDFs.

We are sorry that we did not fully address the reviewers comments regarding the scatter plots appropriately. We added pairwise scatter plots for all precipitation datasets and the three drought years 2005, 2010, 2016 to our analysis (Fig. S3-5). We could not find any obvious biases in the datasets apart from some spikes in the ERA 5 and GLDAS dataset.

Second, there are several tricky steps when translating uncertainties in MCWD into uncertainties in vegetation mortality. Aside from the robustness of the drought-mortality relationship as mentioned by me and the other reviewer in the first round, another reason is that the Lewis et al. 2011 relationship was generated using a specific data set, which was then used to transform MCWD from all the other datasets in this study. Isn't it a more fair comparison to first calibrate the drought-mortality relationship in 2005 using each data set? I understand it might be not easy to get the original data and do the same analysis. However, it is easy to linearly 'project' the different data sets onto the space of the data set as used in Lewis et al. 2011 (TRMM or GPCP) in the first point I made above. For example, if MCWD_TRMM = a * MCWD_CHIRPS + b from regression analysis, we can transform CHIRPS MCWD to the TRMM space using the relationship and then calculate the carbon loss. This can help to more clearly explain some of the differences in Fig. 4.

We thank the reviewer for this comment. We like the idea of getting the specific dataset with which the relationship for the 2005 drought was derived. However, we could not get access to the dataset and could also not reach the author of the study. If the reviewer has access to this dataset we are happy to perform such analysis in a follow-up study. While we also like the linear projection idea of the MCWD datasets, we decided to remove the MCWD-AGB analysis (and also figure 4) from this study as reviewer 1 pointed out flaws of our MCWD-AGB estimation.

Instead, we included two evapotranspiration datasets – DOLCE and GLEAM – in our study. We now also investigate the influence of such variable evapotranspiration input to the drought indices and compare it to the widely used ET=100mm per month. We updated the methods, results and discussion parts of the manuscript accordingly.

Finally, as raised by the other reviewer in the first round, I am now wondering about the suggestion of using an ensemble of rainfall data sets in the last paragraph ("We therefore recommend applying several climate (precipitation) datasets as well as drought metrics to account for model uncertainty when assessing the spatial extent, duration, and location of droughts"). Ensemble arises from the climate systems being chaotic and applies mainly for future predictions. However, for the drought that has already happened, there was a real and single number of

rainfall for each location. So, shouldn't a recommendation of calibrating the gridded data with more ground observations be more logical?

We are sorry that our recommendation causes confusion. We acknowledge the reviewers' conclusion leading to their recommendation of including more ground observation into the dataset. However, we still think that our recommendation using multiple datasets/datasource is valid. Recent studies assessing the impact of drought events e.g. on forests often also use only one dataset to estimate drought extent and severity for both present and past drought events. With our analysis we show that such drought impacts are very dependent on the choice of precipitation dataset, the drought indicator and the evapotranspiration estimate. We think that any study that estimates basin wide drought stress should therefore take multiple datasets, etc into account.

---

## Author Response (AR3)

Reviewer 1

Iam happy to see the revisions, which make the paper more comprehensive and robust. Interesting to see the scatter plots of the precipitation data and how low (!) the spatial correlation is among different rainfall data sets. One thing I note is that WFDEI/GSWP/GPCC seems to have high spatial similarity while others differ in different ways. This can be a useful message in the discussion/conclusion.

On using ensemble of data sets, I agree with the current statement that it is useful to quantify uncertainty. Meanwhile, another piece of information emerging from this analysis is that we really need benchmarking/new constraint given the heterogeneity. My thinking is that unlike future climate projections, the precipitation has already happened (one true value at one place-time combination). So your results are strong evidence that we really don't know what had happened and need to collect/assimilate more ground data in the future. Anyway, not contradictive but instead complementary opinions on the conclusion.

I also like adding DOLCE and GLEAM ET data. Very useful! I think this study would be of interest of many readers.

- *We thank the reviewer for his/her constructive critique, feedback, and ideas that to lead to this final version of the manuscript.*

Reviewer 2

I thank the authors for addressing my concerns. I think this manuscript is looking good, and really highlights the core issues around discrepant drought estimates. I have a few minor comments (see below) and no major comments or criticisms. I hope to see it published. There are some interesting results here that I think will be cited in the years to come.

- *We thank the reviewer for expressing his/her concerns, the constructive feedback, and additional ideas throughout the review process.*

* The last sentence of the abstract is a bit awkward and doesn't say much. Perhaps it would be more impactful to highlight that spatial extent and intensity of even well known droughts is conditional upon the drought metric, and data sources used to calculate them. Or maybe highlight the potential danger of relying on just one data-source, especially if it's one that is not robust for the region.

- *We agree with the reviewer that this sentence does not say much and rephrased it according to the reviewer's suggestion.*

* Section 2.3. Just to be clear, the standard deviation of MCWD is calculated for each individual grid cell, correct? Assuming this is the case (as it should be for making a relative drought metric), I don't understand how absolute values are then derived from the CDF in Fig S1. I would have thought that the absolute values should correspond to the -0.5, -2, and -2.5 standard deviations of MCWD that are spatially explicit. Also, the CWD or MCWD is not normally distributed (it's truncated at 0), so I think the Gaussian CDF was not the right choice. The Gamma CDF might be better, but I don't think it's necessary. Why not just replace Fig S1 with a map of the standard deviation of the absolute MCWD?

- *The reviewer is correct; we indeed normalized the values for rMCWD per gridcell. It is also correct that CWD and MCWD are not normally distributed because of the truncation at 0.  We, however, did not apply any fitting to the MCWD values but only to the anomalies of the MCWD values which do not have this truncation problem at 0. We fit a Gaussian CDF through the empirical distribution across all gridcells and for each drought indicator. By doing so we can compare the CDFs and derive similar classifications between aMCWD and rMCWD. Obviously, we were not clear enough and extended our explanation of this procedure in Methods S1.*

* Section 2.3 about the calculation of (M)CWD should mention that it is reset once per year. It would also be good to mention the details of how it is reset. The wettest month of the year?
- *We added a statement regarding the resetting of MCWD in line 171.*

* Section 2.3.2: A supplement figure mapping the standard deviations used to relativize the RAI would be useful.
- *The distributions of the RAI can be seen in Figure S1c.*

\* Fig. 4: This is a good addition.

- *We are happy that the reviewer likes our new analysis and the corresponding figure 4.*

\* Figs. S3-5: These figures are nice, but I can't tell which product is on the x/y axis.

- *We added the labels to the x/y axis.*

\* typo on p16: " So do CRUNCEP"

- *Fixed*